# Position: Evaluating LLMs in Finance Requires Explicit Bias Consideration

Yaxuan Kong [1 2 *]  Hoyoung Lee [3 *]  Yoontae Hwang [4 *]  Alejandro Lopez-Lira [5]  Bradford Levy [6]
Dhagash Mehta [7]  Qingsong Wen [8 1]  Chanyeol Choi [9]  Yongjae Lee [3]  Stefan Zohren [1]

## Abstract

Large Language Models (LLMs) are increasingly integrated into financial workflows, but evaluation practice has not kept up. Finance-specific biases can inflate performance, contaminate backtests, and make reported results useless for any deployment claim. We identify five recurring biases in financial LLM applications. They include look-ahead bias, survivorship bias, narrative bias, objective bias, and cost bias. These biases break financial tasks in distinct ways and they often compound to create an illusion of validity. We reviewed 164 papers from 2023 to 2025 and found that no single bias is discussed in more than 28 percent of studies. This position paper argues that **bias in financial LLM systems requires explicit attention and that structural validity should be enforced before any result is used to support a deployment claim.** We propose a Structural Validity Framework and an evaluation checklist with minimal requirements for bias diagnosis and future system design. The material is available at `https://github.com/Eleanorkong/Awesome-Financial-LLM-Bias-Mitigation`.

## 1. Introduction

Large Language Models (LLMs) are becoming a common component of modern financial workflows. They read unstructured text, connect information across sources, and

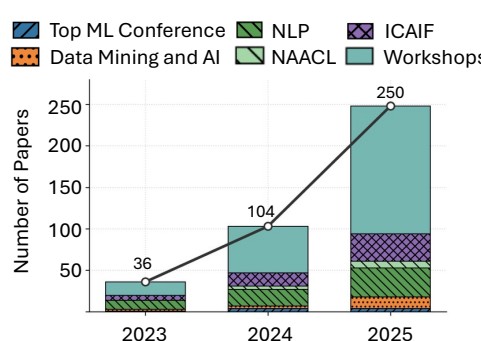

*Figure 1.* Trends in Financial LLM research (2023–2025). Venue breakdown includes Top ML (ICML, ICLR, NeurIPS), Data Mining (KDD, AAAI, IJCAI, CIKM), NLP (ACL, EMNLP), NAACL, ICAIF (financial ML), and workshops; the line shows annual totals. NAACL is listed separately due to conference scheduling.

produce coherent analysis. Prior work reports promising results on financial time series forecasting (Kong et al., 2025; Hwang et al., 2025), risk assessment (Jajoo et al., 2025; Flanagan et al., 2025), regulatory compliance (Kostrzewa et al., 2025; Obiefuna et al., 2025), portfolio construction (Lee et al., 2025a; Yuksel & Sawaf, 2025; Lee et al., 2025c), and information extraction from corporate disclosures and news (Choi et al., 2025b; Dong et al., 2024). This early progress has attracted both academia and industry attention. Between 2023 and 2025, the number of LLM-for-finance papers published in major ML and NLP venues rose from 36 to 250, a 594% increase (6.9×), as shown in Figure 1. We expect the count to keep rising in 2026 as the field expands further. The 2025 NVIDIA State of AI in Financial Services survey reports that the share of respondents using generative AI rose from 40% in 2023 to 52% in 2024. It also reports that generative AI is now used or assessed by over half of respondents, alongside increasing attention to LLM workloads and operational use cases such as document processing and risk management. (NVIDIA, 2025). The pace is fast enough that weak evaluation practice can move from papers into production with almost no friction.

This growth raises a basic question. ***Can LLMs support financial decision making without finance-aware safeguards?*** On this question, we take a hard line. In finance, bias is not a cosmetic issue. Certain evaluation mistakes change the estimand and can make the reported result useless. A backtest that uses future information is invalid.

---

*Equal contribution  [1]University of Oxford, United Kingdom  [2]VulpiVox Intelligence, United Kingdom  [3]Ulsan National Institute of Science and Technology, Ulsan, Republic of Korea  [4]Pusan National University, Busan, Republic of Korea  [5]University of Florida, Florida, United States  [6]University of Chicago Booth School of Business, Chicago, United States  [7]BlackRock, New York, United States  [8]Squirrel Ai Learning, United States  [9]LinqAlpha, United States. Correspondence to: Yongjae Lee <yongjaelee@unist.ac.kr>, Stefan Zohren <stefan.zohren@eng.ox.ac.uk>.

*Proceedings of the 43rd International Conference on Machine Learning*, Seoul, South Korea. PMLR 306, 2026. Copyright 2026 by the author(s).

A benchmark that excludes delisted firms cannot support claims about real-world risk. An evaluation that rewards fluent stories and confident guesses can look strong while measuring persuasion rather than knowledge. A study that ignores trading frictions, inference cost, and latency can turn an uneconomic system into a winner on paper. We focus on five recurring failure modes that we call the five sins. Look-ahead bias occurs when the system uses information that was not available at decision time, through model weights or external context. Survivorship bias occurs when the evaluation universe silently drops firms that merge, delist, or fail. Narrative bias occurs when the model produces a clean rationale that is not supported by the evidence available at the time. Objective bias occurs when training and alignment reward confident completion instead of calibrated uncertainty and safe refusal. Cost bias occurs when studies report gross performance while assuming zero execution friction and zero operating cost. These sins often stack. They create an illusion of validity where strong numbers coexist with invalid backtests and systems that cannot be deployed.

The literature does not treat these issues with the seriousness they deserve. Among the 164 main conference papers we reviewed from 2023 to 2025 (Figure 2)[1], only 26.8% acknowledge look-ahead bias, and survivorship bias appears in just 1.2% of studies. This is not a minor reporting gap. It means many published numbers do not support the claims that readers and practitioners want to make from them. Our study points to the same failure. We surveyed 112 researchers and practitioners. Among the 50 respondents who completed all required questions, 74% reported that ready-to-use evaluation tools are scarce or non-existent, and 50% identified the lack of tools and frameworks as the biggest bottleneck to mitigation. The gap presents differently across groups: academics often recognize the biases' names but struggle to diagnose mechanisms, while industry participants recognize the consequences but lack standardized checks.

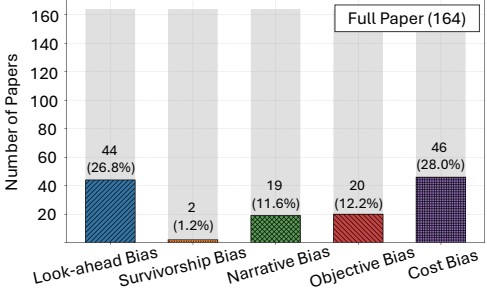

*Figure 2.* Bias distribution across 164 LLM-for-Finance papers. Gray bars denote total paper; colored denote mentions per bias.

Given the gap between risks and current practice, **we argue**

[1]Our review includes main conference papers from leading venues in ML (ICML, ICLR, NeurIPS), Data Mining and AI (KDD, AAAI, IJCAI, CIKM), NLP (ACL, EMNLP, NAACL), and the financial AI conference ICAIF.

**that bias in financial LLM systems requires explicit attention and that structural validity should be enforced before any result is used to support a deployment claim.** This paper makes two claims. First, the field needs to stop treating generic language evaluation as a proxy for financial validity. Second, the field needs diagnostics that reviewers and practitioners can apply without guesswork. To address these needs, we propose a Structural Validity Framework with an evaluation checklist that treats core requirements as pass or fail. The framework comprises five components, each targeting one of the sins identified above. Temporal sanitation enforces non-anticipativity in model weights, retrieval, and tools. Dynamic universe construction prevents survivor-conditioned evaluation. Rationale robustness treats explanations as testable objects rather than decoration. Epistemic calibration makes uncertainty and abstention part of the interface and the score. Realistic implementation constraints force reporting of net utility under costs and latency. We use this framework to link each sin to the tasks where it is most likely to appear and to propose mitigations.

## 2. Biases That Are Being Overlooked

As LLMs are increasingly used in quantitative finance, their evaluation is often based on protocols that contain implicit assumptions. These assumptions can systematically inflate reported performance and lead to misleading conclusions. In this section, we identify five recurring pitfalls, referred to as the "five sins", that can distort the assessment of a Financial LLM-based system $M$. As illustrated in Figure 3 and Table 1, these biases can collectively create an illusion of validity, where apparently strong empirical results conceal fundamental weaknesses in the evaluation design.

### 2.1. Sin #1: Look-Ahead Bias

Look-ahead bias arises when an evaluation uses information that did not exist, or was not accessible, at historical time $t$. A valid decision rule induced by system $M$ must depend only on the information set $\mathcal{I}_t$ available at time $t$. This is the non-anticipativity condition. In financial LLM systems, violations typically occur through two channels: **(1)** parametric knowledge embedded in model weights, and **(2)** external context introduced through retrieval-augmented generation (RAG), tools, or live data sources.

**Issue 1: Parametric Knowledge Leakage.** Let $M$ be pre-trained on corpora collected up to calendar time $T_{\text{cutoff}}$. When backtesting at a historical time $t < T_{\text{cutoff}}$, the model is susceptible to *time travel*, leveraging parametric knowledge of future information to compromise the integrity of the evaluation (Golchin & Surdeanu, 2023; Sarkar & Vafa, 2024; Lopez-Lira et al., 2025; Ni et al., 2025; Gao et al., 2025). Consequently, the effective information set exceeds $\mathcal{I}_t$ (the information set available at time $t$). However, the

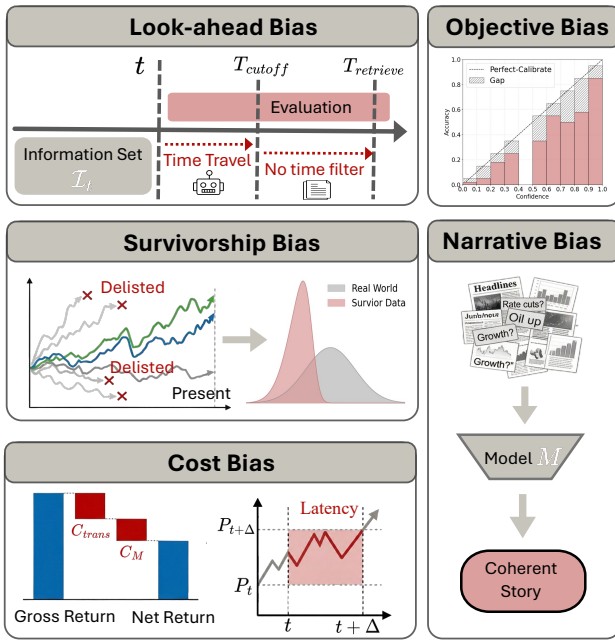

*Figure 3.* The illusion of validity in Financial LLM evaluation. The figure illustrates five common biases that arise from data construction, model behavior, and deployment assumptions.

mere existence of a stated knowledge cutoff does not guarantee the exclusion of post-cutoff information. As noted by Paleka et al. (2025), relying solely on nominal cutoffs is insufficient. Empirical evidence reveals instances where models produce specific information about events occurring after their stated cutoff dates, suggesting that temporal knowledge boundaries remain opaque (Golchin & Surdeanu, 2023; Cheng et al., 2024). Given that closed-source models often lack transparency regarding system prompts and post-training updates, validating performance on open-source models is essential to ensure reliability.

**Issue 2: External Knowledge Leakage.** A second source of look-ahead bias arises when a system $M$ integrates external context through RAG. At simulation time $t$, the system should retrieve context $c_t$ from a retriever $\mathcal{R}$ restricted strictly to the $\mathcal{I}_t$ in response to a query $q_t$:

$$c_t \sim \mathcal{R}(\mathcal{I}_t, q_t) \qquad (1)$$

In practice, this assumption often fails. The problem becomes more severe when retrieval relies on live search engines or on documents that remain editable after publication. Modern ranking systems rely on signals such as clicks, links, and retrospectively assessed importance. These signals were not observable at time $t$. As shown by Paleka et al. (2025), searches restricted to pre-2020 documents can still surface content shaped by the 2021 U.S. Capitol events. Alur et al. (2025) further note that many documents change over time while retaining their original publication dates. As a result, retrieval may return updated pages or live data that bypass temporal filters.

This behavior causes the retrieved information set to exceed the true information set $\mathcal{I}_t$. The model then appears to act on information refined by future aggregation and validation. Performance metrics under these conditions do not estimate the returns of non-anticipative strategies. They instead reflect pseudo-PnL from a temporally contaminated environment. Such contamination includes future edits, retrospective curation, and survivorship effects. This distortion becomes more severe as retrieval depth and corpus size increase. To reduce this bias, retrieval should rely on point-in-time data with clear provenance. Suitable sources include archived snapshots and version-controlled corpora with explicit as-of metadata. A screening layer can also detect temporal violations, such as references to post-$t$ events. If an LLM performs this screening, it should follow the same temporal constraints as $M$. Without these safeguards, backtest results reflect temporally inconsistent information flows rather than evidence of alpha.

### 2.2. Sin #2: Survivorship Bias

Survivorship bias is a classic concern in empirical finance where the evaluation universe is restricted to entities surviving the sample period (Brown et al., 1992; Rohleder et al., 2011). This bias typically manifests when delisted firms are excluded. Let $S_t \in {0, 1}$ denote whether an entity remains active at time $t$, such as listing status. A valid evaluation targets the population distribution $P(X)$ over market observations $X_{i,t}$. Common dataset construction instead relies on the conditional distribution $P(X \mid S_t = 1)$. This restriction distorts evaluation through four channels: **(1)** question generation, **(2)** model behavior, **(3)** data distributions, and **(4)** benchmark outcomes.

**Issue 1: Bias in Ex-post Question Generation.** Survivorship bias becomes acute when benchmark questions are generated retrospectively from future information (Dai et al., 2024; Paleka et al., 2024). Because media coverage inherently correlates with ongoing corporate activity and investor attention, firms that fail or quietly exit the market tend to be underrepresented (Paleka et al., 2025). Consequently, question-generation algorithms that sample by news volume rarely produce queries about delisted or bankrupt firms, effectively excluding negative cases. This is problematic because forecasting distress and failure is central to real-world risk management, yet it is precisely the regime that disappears under survivor-conditioned sampling.

**Issue 2: Intrinsic Model Bias.** Survivorship bias in the underlying corpora can also translate into systematic model preferences. If the pre-training text disproportionately reflects large, well-known, and historically surviving firms, then the learned associations approximate a conditional view of the world closer to $p(x \mid S = 1)$. Consistent with this, Lee et al. (2025b) report that LLMs tend to favor large-cap

*Table 1.* Five overlooked biases in financial LLM evaluation with definitions, examples, common mitigation strategies, and limitations.

| Bias Type | Definition | Financial Example | Common Solution | Why the Solution Fails |
|---|---|---|---|---|
| **Look-Ahead Bias** | The model accidentally uses future information that would not have been known at the time a decision was supposed to be made. | A 2018 trading backtest benefits from knowledge of post-2020 macro events embedded in model weights or retrieved from updated web sources. | (1) Declare a knowledge cutoff date (2) Filter retrieved documents by publication time | (1) Cutoffs are opaque and not verifiable (2) Retrieved documents are edited and re-ranked using future information |
| **Survivorship Bias** | The evaluation only includes companies that survived, while failed or delisted firms are silently excluded. | Benchmarks focus on currently listed firms, while bankrupt companies rarely appear in questions or datasets. | (1) Use standard historical market datasets (2) Build benchmarks retrospectively | (1) Failure cases are dropped (2) Downside risk and tail events are under-represented |
| **Narrative Bias** | The model tells a convincing story that sounds reasonable but is not fully supported by the data. | An earnings-call summary presents a clean growth story while ignoring scattered warning signs. | (1) Rely on fluent summaries (2) Use Chain-of-Thought explanations | (1) Coherence hides uncertainty and conflicting evidence (2) Stories break under regime shifts |
| **Objective Bias** | The model is trained to appear confident rather than express uncertainty or admit ignorance. | A model confidently issues a buy or sell recommendation despite weak or ambiguous evidence. | (1) Apply instruction-tuned LLMs directly (2) Evaluate using accuracy or preference scores | (1) Training rewards confident guessing (2) Miscalibration and hallucinations persist |
| **Cost Bias** | Performance looks good on paper because real-world costs and delays are ignored. | A complex LLM trading system is profitable in backtests but loses money once inference costs and latency are included. | (1) Assume zero inference cost (2) Assume instantaneous execution | (1) Costs compound over time (2) Latency causes slippage and signal decay |

stocks under conflicting evidence. Similarly, Dimino et al. (2025b) observe that firm size consistently increases model confidence. Ultimately, this deceptive robustness reinforces historical dominance, posing a significant threat to fairness.

**Issue 3: Skewed Data Distributions.** Financial data typically exhibit heavy-tailed distributions driven by rare but catastrophic events. In most cases, conditioning on survival can attenuate the left tail by down-weighting failure-associated trajectories. A convenient way to express this is as mixtures of surviving and non-surviving components:

$$p_{\text{world}}(X) = p(X \mid S = 1)P(S = 1) \\ + p(X \mid S = 0)P(S = 0). \tag{2}$$

If the dataset drops the $S = 0$ components, then models are trained on an environment that systematically underrepresents crisis-linked features. This can inflate apparent stability and reduce true tail risk awareness.

**Issue 4: Distorted Benchmarks and Backtesting.** Survivorship bias induces systematic overestimation in benchmark evaluation and financial backtesting. Many benchmarks are constructed retrospectively based on currently surviving entities ($S = 1$). This practice causes performance metrics $\mathcal{B}(M)$, such as question-answering accuracy for surviving firms or portfolio returns, to overestimate true population performance. When non-surviving entities exhibit lower performance than surviving ones, the expected value over the survivor dataset $\mathcal{D}_{\text{survive}}$ necessarily exceeds that over the true market distribution $\mathcal{D}_{\text{true}}$:

$$\mathbb{E}[\mathcal{B}(M) \mid \mathcal{D}_{\text{survive}}] > \mathbb{E}[\mathcal{B}(M) \mid \mathcal{D}_{\text{true}}]. \tag{3}$$

This provides a theoretical basis for why backtest returns appear inflated when real-world market drag factors are ignored. To address these distortions, benchmarks and backtesting procedures should operate within universes that account for survivorship bias. This approach relies on dynamic universes that include all entities present at evaluation time $t$, including those subsequently delisted. Such design mitigates bias and reflects genuine market uncertainty rather than retrospectively curated distributions.

### 2.3. Sin #3: Narrative Bias

Narrative bias refers to a recurring failure mode in which LLMs generate coherent financial explanations whose factual claims, causal links, or risk implications are not sufficiently justified by the evidence available at the decision time. Because LLMs are trained to optimize next token prediction (Brown et al., 2020), they are rewarded for producing fluent and internally consistent text rather than for faithfully representing uncertainty or conflicting signals (Bubeck et al., 2023; Bachmann & Nagarajan, 2024). Consequently, when applied to fragmented and noisy financial data, this objective can induce models to stitch disparate signals into unified stories that may not reflect the underlying evidence. The concern is therefore not merely that a rationale sounds persuasive, but that fluency can mask missing citations, omitted counter-evidence, factual errors, or unjustified causal interpretation. We treat narrative bias as an evidence-grounding and auditability problem, rather than as a subjective judgment of whether a story appears plausible.

**Issue 1: Bias in Information Processing.** Narrative bias is especially consequential in unstructured data processing tasks such as earnings call digests, analyst-note synthesis, and risk-report drafting. Because LLMs are trained to optimize linguistic plausibility and narrative coherence rather than factual precision, they may reshape source material in ways that alter its informational content. For example, Alessa et al. (2025) show that LLMs exhibit framing bias during summarization, particularly as a tendency to shift neutral or negative text toward more positive sentiment. In finance, such shifts can suppress weak but important distress signals or over-emphasize a single "main story" at the expense of conflicting evidence. As a result, users may receive summaries that are easier to act on but less faithful to the underlying record, which can bias risk perception and decision thresholds.

**Issue 2: Bias in Inference.** A fundamental limitation in investment reasoning is that the training objective of LLMs does not ensure the identification of robust drivers of returns. Furthermore, Chain-of-Thought (CoT) rationales should not be treated as transparent traces of the model's internal decision process. For example, Turpin et al. (2023) show that CoT can rationalize outputs post hoc and is sensitive to non-semantic factors, while Zhao et al. (2025a) argue that CoT-style reasoning can degrade under distribution shift. Given that financial markets are inherently non-stationary, narratives overfitted to historical data patterns will inevitably lose causal explanatory power when confronted with regime shifts. Therefore, we caution against treating strong back-tests or fluent reasoning as evidence of causal understanding, unless the rationale is grounded in time-valid evidence and strictly audited for unsupported claims or biased confidence.

## 2.4. Sin #4: Objective Bias

Objective bias describes a fundamental misalignment between the objectives optimized during LLM training and the requirements of the financial domain. While financial decision-making prioritizes risk management and compliance, standard LLMs are primarily optimized for likelihood maximization and aligned via human preference (Ouyang et al., 2022). This discrepancy creates a structural tendency to favor plausible-sounding text over reliable uncertainty quantification or adherence to safety norms. As a result, outputs may appear confident and coherent while obscuring risk exposure or compliance violations. We characterize this failure mode through two channels: **(1)** incentive-induced miscalibration and **(2)** alignment mismatch.

**Issue 1: Incentive-Induced Miscalibration.** Objective bias primarily appears as calibration failure induced by the model's incentive structure. LLMs are trained to maximize likelihood and are often aligned through preference optimization. These objectives encourage verbal overconfidence under biased reward models (Leng et al., 2024; Jiang et al., 2025). Confident and specific answers receive higher reward than expressions of ignorance or ambiguity. As argued by Kalai et al. (2025), training regimes that penalize abstention statistically force models to hallucinate. The model guesses with high confidence once queries exceed its reliable knowledge. In finance, this produces excessive or uninformative certainty (Liu et al., 2024). The model hides ignorance behind fluent prose rather than signaling low confidence. Evidence from prediction markets confirms weak calibration in future event prediction (Yang et al., 2025a).

**Issue 2: Alignment Mismatch.** A deeper structural risk arises because generic alignment techniques, such as RLHF, do not encode the safety and compliance constraints required in high-stakes domains. Standard alignment aims to make models helpful and harmless in a general sense. It often defines helpfulness as the direct satisfaction of user requests. In finance, however, safety requires strict adherence to regulatory norms and risk-aware communication. LLMs may prioritize user satisfaction and produce persuasive but speculative investment advice (Ding et al., 2025). As Lo & Ross (2024) note, this mismatch creates a hazard. The model's optimized behavior diverges from ethical and safety standards for responsible financial decisions. This misalignment leads to critical risk where users prioritize persona over accuracy (Takayanagi et al., 2025; Qu et al., 2025).

## 2.5. Sin #5: Cost Bias

We define cost bias as the gap between performance reported under idealized evaluation assumptions and net utility achievable under deployment constraints. Much academic work (Laskar et al., 2024) implicitly treats a system $M$ as free to run and instantaneous. In practice, systems $M$ often rely on multi-step prompting and nontrivial serving infrastructure, which incur high cost and latency. Ignoring these factors can favor heavier pipelines that score well on gross metrics but perform poorly in deployment. Below, we decompose cost bias into **(1)** monetary cost and **(2)** latency.

**Issue 1: Monetary Cost Bias.** Monetary cost bias arises from the systematic omission of explicit operational expenditures from evaluation metrics. For financial tasks, the economically relevant quantity is not gross performance $R_{\text{gross}}$ but rather the net return after deducting transaction costs $C_{\text{trans}}$ and model inference and operational costs $C_M$:

$$R_{\text{net}} = R_{\text{gross}} - C_{\text{trans}} - C_M \qquad (4)$$

However, prevailing evaluation practices implicitly assume $C_{\text{trans}} = C_M = 0$ and optimize exclusively for $R_{\text{gross}}$. Sophisticated models with elaborate prompt chains, multi-step reasoning, or frequent tool use may appear superior to simple baselines, despite much higher operational costs. In deployment settings, accumulated $C_M$ can erode economic value. This issue extends beyond trading to generative AI systems with non-negligible inference costs.

**Issue 2: Latency Cost Bias.** Latency cost bias occurs when evaluations assume zero delay between observation at time $t$ and action. Standard evaluation frameworks treat action execution as instantaneous once information arrives. In practice, $M$ incurs generation latency $\Delta_{\text{gen}}$ from multi-step inference, RAG, and other computation. In trading, this latency causes slippage, since orders execute at $P_{t+\Delta_{\text{gen}}}$ instead of $P_t$. In other time-sensitive tasks, such as real-time news analysis or report generation, latency reduces value as the information advantage decays over time. Performance reported under an implicit $\Delta_{\text{gen}}$ assumption can therefore mislead, especially for short-horizon strategies.

**Call to Action:** *Our analysis reveals that all five biases are consistently under-reported, with no single bias mentioned in more than 28% of the 164 papers surveyed. We advocate explicit attention to these biases, which should not be overlooked in the evaluation of financial LLMs.*

## 3. The Structural Validity Framework: Guidance for Evaluation

Progress in Financial LLM research has outpaced the evaluation standards required for credible inference. Most language evaluation metrics assume fixed inputs, fixed targets, and an evaluator that can ignore feasibility. Financial systems violate these assumptions because they implement policies that condition on evolving information sets, operate over time-varying tradable universes, and incur execution frictions and operational costs.

We propose a **structural validity framework** that establishes minimum requirements for interpreting a reported backtest as evidence of deployment performance, rather than as an artifact of leakage, dataset curation, overconfident guessing, persuasive rationales, or frictionless simulation. We treat the requirements below as binary diagnostics: if any requirement fails, the reported result can only support a stress test interpretation or a proof of concept, and it cannot support a claim of deployable alpha. Figure 4 presents an overview of the structural validity checklist. Appendix C provides the details, with an interactive version available.

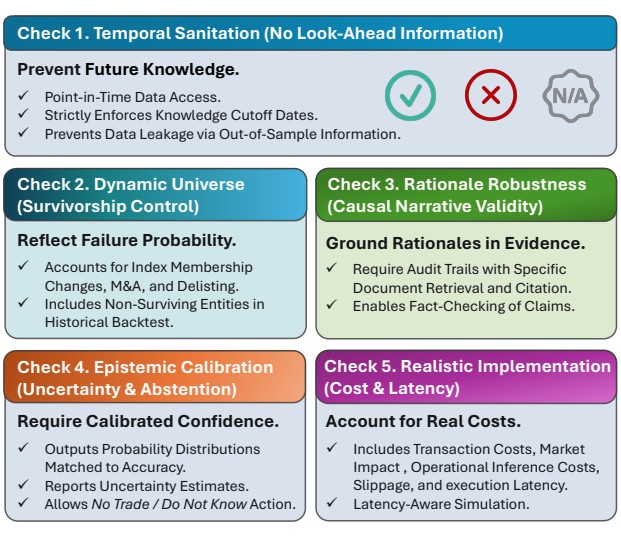

*Figure 4.* Overview of the Structural Validity Checklist.

### 3.1. Temporal Sanitation

Temporal sanitation enforces non-anticipativity at the level of each decision time $t$. **The system is required to draw only on information that existed and was accessible at $t$,** whether that information enters through model weights, re-

trieval, or tool outputs. We mandate that authors disclose the latest calendar date used for pre-training, as well as in any subsequent fine tuning or instruction tuning that produced the deployed weights. We also require that the evaluation window $[t_0, t_1]$ respect that disclosure.

We extend this requirement to external context. Retrieval corpora and indices should be constructed from documents available at each simulated time $t$, and the system should operate on archived snapshots with verifiable as-of timestamps when the underlying source is mutable. Evaluation protocols that query present-day search engines or knowledge bases assembled after $t$ are prohibited unless they return point-in-time material and the study documents that guarantee. Ranking and curation effects should also be controlled: indices and rankers should avoid signals incorporating later attention, such as engagement data, link graphs, or metadata reflecting outcomes after the evaluation period.

Structured market and fundamental data should also be point-in-time series. If a dataset has been revised or cleaned using information unavailable at $t$, authors should explicitly model the revision process or treat the data as contaminated. Finally, we require comprehensive trace documentation. Evaluations should record prompts, retrieved documents or identifiers, tool outputs, and decision timestamps. Where full trace logging is infeasible due to licensing or access constraints, studies should document data sources, versioning, as-of metadata, and disclosure limitations. This record should allow an independent auditor to verify temporal consistency at the level of individual actions.

### 3.2. Dynamic Universe Construction

Dynamic universe construction prevents survivor conditioned evaluation. **The study should define a time-indexed tradable universe $\mathcal{U}_t$ for each decision time $t$, and draw task sampling and trade simulation from $\mathcal{U}_t$ rather than a universe defined at the end of the sample.** Authors should include entities that delist, merge, or fail, provided they were observable at $t$. Dropping these cases fails structural validity as it removes regimes that dominate risk.

When evaluations include question answering or event driven prompts, universe construction affects benchmark generation. We require sampling rules that do not proxy survival by selecting entities in proportion to ex post news volume or present day prominence without correction. The benchmark should contain negative cases that represent distress and failure regimes, not only continuing success stories. We also mandate basic universe diagnostics. Authors should report the fraction of observations that come from entities that delist within the sample, and they should report outcome distributions separately for surviving and non-surviving entities. A benchmark with almost no non-surviving trajectories cannot support claims about performance in real markets.

### 3.3. Rationale Robustness

Rationale robustness addresses the risk that fluent explanations substitute for evidence. **Evaluators should treat model rationales as testable objects and ensure that, for each decision at time $t$, rationales rely only on temporally sanitized sources available at $t$.** Key factual claims should trace to specific retrieved passages or structured fields, with traces stored in evaluation logs. Evaluators should audit factual consistency to detect references to nonexistent events, incorrect quantities, or information that appears after $t$. Authors should report violation rates alongside performance metrics rather than relying on selected examples. Evaluations should also include insufficiency tests and negative controls. These tests should verify that the system does not generate detailed causal stories or confident rationales and does not make trades when the available evidence is absent, irrelevant, or scrambled.

### 3.4. Epistemic Calibration

Epistemic calibration aligns model output with the needs of financial decision support. Standard token prediction and preference tuning push models toward confident completions even when the query lies outside the model's knowledge region. An evaluation that forces the model to guess converts ignorance into false precision and rewards fluent certainty. Interfaces and scoring rules should therefore make uncertainty observable and treat abstention as a legitimate outcome. The system should output a predictive distribution, a calibrated confidence score, or a structured uncertainty estimate that downstream risk controllers can consume. **The action space should include explicit abstention, such as a "No Trade" or "Do Not Know" option, and the evaluator should score abstention as a valid result rather than an automatic failure.** This design prevents evaluation protocols from converting ignorance into false precision.

### 3.5. Realistic Implementation Constraints

Realistic evaluations should account for execution frictions and operating costs. Reported results should reflect net utility under the latency and cost structure of the deployed agent, as assumptions of instantaneous action and zero cost evaluate a frictionless abstraction rather than the proposed system. Implementation constraints should also form part of the objective. Authors should measure the full distribution of $\Delta_{\text{gen}}$ under hardware, concurrency, and tool-usage conditions, as reporting a single best-case latency does not support deployment claims. Backtests should execute trades at $P_{t+\Delta_{\text{gen}}}$ or under an equivalent slippage model. Studies should specify spread, fees, and market impact assumptions; report token usage and tool calls; and incorporate operational costs into $C_M$. Net metrics should serve as primary outcomes, and comparisons should use budget-matched baselines with similar latency and cost constraints. **Without budget matching, evaluations may favor heavier pipelines that improve gross metrics while degrading net utility.**

## 4. The Need for a Structural Validity Framework: Evidence from a User Study

Without structural validity, financial LLM evaluation becomes unreliable due to data leakage and hidden biases. **The Structural Validity Framework** defines clear pass/fail prerequisites that address these risks through temporal sanitation, dynamic universe construction, epistemic calibration, rationale robustness, and realistic implementation constraints. The framework enables fair comparison under realistic constraints and establishes a concrete standard for methodological rigor. We therefore present a user study that validates the checklist and demonstrates the challenges discussed in Sections 2 and 3.

**User study.** We conducted a user study with 112 practitioners and researchers in financial LLM applications to assess participant backgrounds, bias awareness, and current evaluation practices. Among 50 respondents who completed all required questions, 48% reported that current LLM-for-finance research provides only "to a small extent" clear guidance for identifying or measuring biases (Q8), while 74% indicated that ready-to-use evaluation tools are either "scarce" (48%) or "non-existent" (26%) (Q9), and 50% identified "lack of evaluation tools/frameworks" as the biggest bottleneck to effective bias mitigation (Q11) (see Figure 5). These results demonstrate that the absence of standardized evaluation frameworks is a practical barrier preventing rigorous evaluation in the field, underscoring the urgent need for a structural validity framework. The detailed findings and complete analysis can be found in Appendix A.

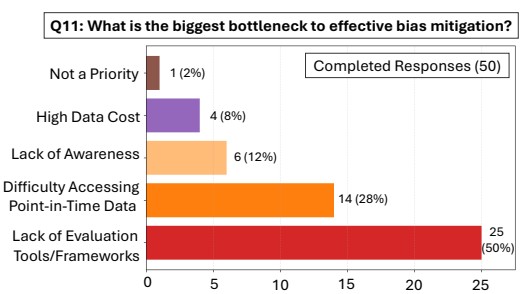

*Figure 5.* Biggest bottlenecks to effective bias mitigation (Q11).

**Call to Action:** *Our user study reveals that 74% of practitioners lack adequate tools for bias evaluation, with 50% identifying this as the primary bottleneck. We advocate the use of Structural Validity Checklist (Appendix C) to enable bias diagnosis and reproducible evaluation.*

## 5. Alternative Views

**First**, some may contend that requiring the checklist at early research stages slows innovation. They may argue that early work should prioritize whether $M$ shows meaningful capability on financial tasks rather than deployment oriented constraints. From this view, postponing requirements such as point-in-time sanitation, dynamic universe construction, and explicit cost accounting avoids discouraging exploratory studies that may later mature into rigorous systems.

**Second**, critics may argue that the framework raises equity concerns because the data required for valid evaluation is costly and often proprietary. They may claim that reliance on point-in-time fundamentals, delisting histories, archived disclosures, and realistic execution data may create substantial access and cost barriers for smaller academic groups and independent researchers relative to well-funded institutions.

**Third**, some may argue that evaluating LLMs under strict financial constraints mischaracterizes them as specialized trading systems rather than general-purpose reasoning engines. Requirements such as tradable universes, execution frictions, and latency primarily test pipeline engineering rather than reasoning ability, and a framework focused on deployment feasibility may reduce $M$ to a trading tool instead of a broadly capable model for general reasoning.

**Fourth**, some may argue that hallucination consists of localized, detectable, and often reversible errors, and therefore poses a more immediate concern than bias. From this perspective, prioritizing bias evaluation is seen as misplaced when hallucination errors more directly affect correctness and reliability, and when such errors can be addressed through improved training or prompt design.

While these counterarguments raise relevant considerations, they do not weaken the need for the **Structural Validity Framework** in claims about deployable performance. If temporal sanitation fails, the induced policy $T(M)$ uses information unavailable at time $t$, making it infeasible. Calling such results "early stage" does not fix the inference problem, since the estimand is not deployable. The same flaw appears in survivor-conditioned universes and frictionless reporting. Excluding delisted entities removes the regimes that drive downside risk, while omitting costs and latency inflates gross output and obscures net economic utility. Ignoring $C_M$ and $\Delta_{\text{gen}}$ changes the object of measurement and can make an uneconomic system appear effective.

Concerns about access and equity deserve attention, but democratization cannot justify false scientific claims. Hallucination and bias are orthogonal failure modes, with hallucinations localized and correctable, while bias reflects structural defects that distort the estimand and perpetuate inequitable outcomes if left unaddressed. When researchers lack the data required to support a deployable claim, the appropriate response is to narrow the claim itself. Lowering the bar on validity does not democratize science. It democratizes error and makes it harder for academics and practitioners to distinguish real progress from artifacts. **For these reasons, we insist that the framework is not an optional standard that can be postponed without consequence.**

## 6. Related Works

To our knowledge, no standard framework exists for assessing all biases in LLM-based financial applications; we propose the first such guidance. Existing work has largely focused on look-ahead bias due to its prominence in finance. We discuss the related literature throughout the paper (See Section 2); thus, this section provides a brief selection of representative works. A dashboard of recent LLM-for-Finance research is included, with details in Appendix B.

**Bias in LLMs for Finance.** Recent studies document systematic biases in LLM-for-Finance tasks. Lee et al. (2025b) show that LLMs favor large-cap stocks and contrarian strategies and exhibit confirmation bias in investment analysis. Other work identifies representation bias tied to firm characteristics such as size and sector, as well as foreign bias, where U.S.-trained models produce systematically more optimistic forecasts for Chinese firms due to training data asymmetries (Cao et al., 2025). Additional studies report data-snooping biases in LLM-based investment evaluations, leading to inflated backtest performance (Li et al., 2025a).

**LLM Applications in Finance.** The application of LLMs to financial tasks has seen rapid growth. Recent work includes agent-based trading systems (Henning et al., 2025), portfolio optimization methods (Zhao et al., 2025b), and market simulation (Yang et al., 2025b). Sentiment analysis applications (Wei & Liu, 2025) and retrieval-augmented generation systems for financial question answering (Choi et al., 2025a) have also emerged. However, the evaluation of these systems has primarily focused on task performance metrics rather than systematic bias assessment.

## 7. Conclusion

This position paper argues that bias in financial LLM systems requires explicit attention and that structural validity should be enforced before any result is used to support a deployment claim. To support this, we illustrate how these biases arise in practice and introduce the **Structural Validity Framework** for bias diagnosis. Evidence from the literature and our user study suggests that existing work provides limited evaluation guidance. We therefore call on researchers to **(1) identify and discuss these biases** in their work, and **(2) adopt the checklists to support reproducible evaluation**, which enables fairer and more reliable financial LLM-based systems.

## Acknowledgments

This work was supported by the National Research Foundation of Korea (NRF) grant funded by the Korea government (MSIT) (No. RS-2023-00242528 and No. RS-2025-24803208) and the Institute of Information & Communications Technology Planning & Evaluation (IITP) grant funded by the Korea government (MSIT) (No. RS-2020-II201336, Artificial Intelligence Graduate School Program (UNIST)).

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

# A. Motivation - User study

Below provides a comprehensive analysis of the user study conducted with practitioners and researchers in financial LLM applications. The study examined participant backgrounds, bias awareness, and the current state of evaluation practices and tools in the field. We distributed an online survey to 112 industry practitioners and researchers working with LLMs in financial applications. The survey was designed to get the information about: **(1)** Participant professional backgrounds, **(2)** Awareness and familiarity with various biases affecting financial LLM evaluation **(3)** Perceived criticality of different biases **(4)** Current evaluation practices and tool availability **(5)** Barriers to effective bias mitigation. Among the 112 total respondents, 50 participants (44.6%) completed all required questions (Q1–Q11). The following analysis is based on these 50 completed responses. The detailed dashboard can be found at `https://github.com/Eleanorkong/Awesome-Financial-LLM-Bias-Mitigation`.

## A.1. Participant Background

**Question 1: Primary Role.** Figure 6 (Q1) shows the distribution of participants by their primary role. The study included a balanced mix of academic researchers and industry practitioners: 56% (28 participants) identified as Academic Researchers (PhD Students or Academic Professors), while 44% (22 participants) identified as Industry Practitioners (Quants, Data Scientists, Traders, etc.). This ensures that our findings reflect perspectives from both research and practical contexts.

**Question 2: Professional Experience** Figure 6 (Q2) presents the distribution of professional experience in the financial industry. The majority of participants (40%, 20 participants) reported 2–5 years of experience, followed by 28% (14 participants) with less than 2 years of experience. Participants with 10+ years of experience comprised 18% (9 participants), while 14% (7 participants) reported 5–10 years of experience. This distribution indicates that our sample includes both early-career and experienced professionals, providing a comprehensive view of the field.

**Question 3: LLM Usage Frequency** Figure 6 (Q3) illustrates how frequently participants utilize LLMs for quantitative or analytical financial tasks. A majority of 56% (28 participants) reported using LLMs daily, demonstrating high engagement with LLM technologies in their work. An additional 20% (10 participants) use LLMs weekly, and another 20% (10 participants) use them occasionally (monthly or quarterly). Only 4% (2 participants) reported rarely or never using LLMs, indicating that our sample consists primarily of active LLM users in financial contexts.

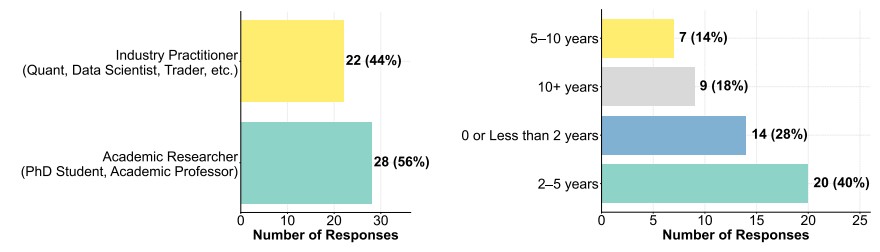 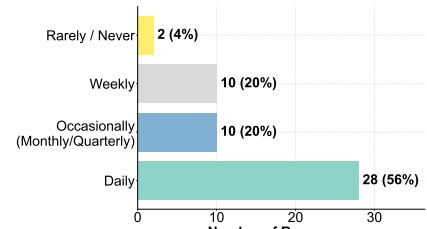

Q1. Which of the following best describes your primary role?

Q2. How many years of professional (work) experience do you have in the financial industry?

Q3. How often do you utilise LLMs for quantitative or analytical financial tasks in your research or professional activities?

*Figure 6.* User study - Participant Background and LLM Usage (Total Response: 50).

## A.2. Bias Awareness and Familiarity

**Question 4: Familiarity with Biases** This question assessed participants' familiarity with five specific biases before taking the survey. Figure 7 (Q4) present the results for each bias type.

(1) **Look-ahead Bias:** As shown in Figure 7 (Q4), look-ahead bias was the most familiar bias among participants, with 44% (22 participants) reporting being "Very familiar" and 18% (9 participants) being "Extremely familiar". Only 2% (1 participant) reported being "Not familiar at all", indicating high awareness of this bias in the community.

(2) **Survivorship Bias:** Figure 7 (Q4) shows that survivorship bias had moderate familiarity, with 28% (14 participants) being "Very familiar" and 12% (6 participants) being "Extremely familiar". However, 16% (8 participants) reported being "Not familiar at all", suggesting lower awareness compared to look-ahead bias.

(3) **Narrative Bias:** As illustrated in Figure 7 (Q4), narrative bias showed the most distributed familiarity levels, with 26% (13 participants) being "Slightly familiar" and 24% (12 participants) being "Moderately familiar". This suggests that narrative bias is less well-known in the community, with 10% (5 participants) reporting no familiarity.

(4) **Objective Bias:** Figure 7 (Q4) indicates that objective bias had moderate awareness, with 32% (16 participants) being "Moderately familiar" and 22% (11 participants) being "Very familiar". Similar to narrative bias, 10% (5 participants) reported no familiarity.

(5) **Cost Bias:** As shown in Figure 7 (Q4), cost bias had the lowest familiarity among the five biases, with 32% (16 participants) being "Moderately familiar" and 30% (15 participants) being "Slightly familiar". Only 8% (4 participants) reported being "Extremely familiar", and 10% (5 participants) had no familiarity, highlighting a significant gap in awareness of operational cost considerations in LLM evaluation.

**Question 5: Frequency of Bias Discussion in Literature** Figure 7 (Q5) shows how often participants see these biases discussed in financial LLM papers or reports. The results reveal a concerning gap: 36% (18 participants) reported seeing biases discussed "Rarely", and 8% (4 participants) reported "Never". Only 20% (10 participants) see biases discussed "Often", and merely 6% (3 participants) see them discussed "Very often". This finding underscores the limited attention given to bias discussion in current literature, which aligns with our paper's motivation.

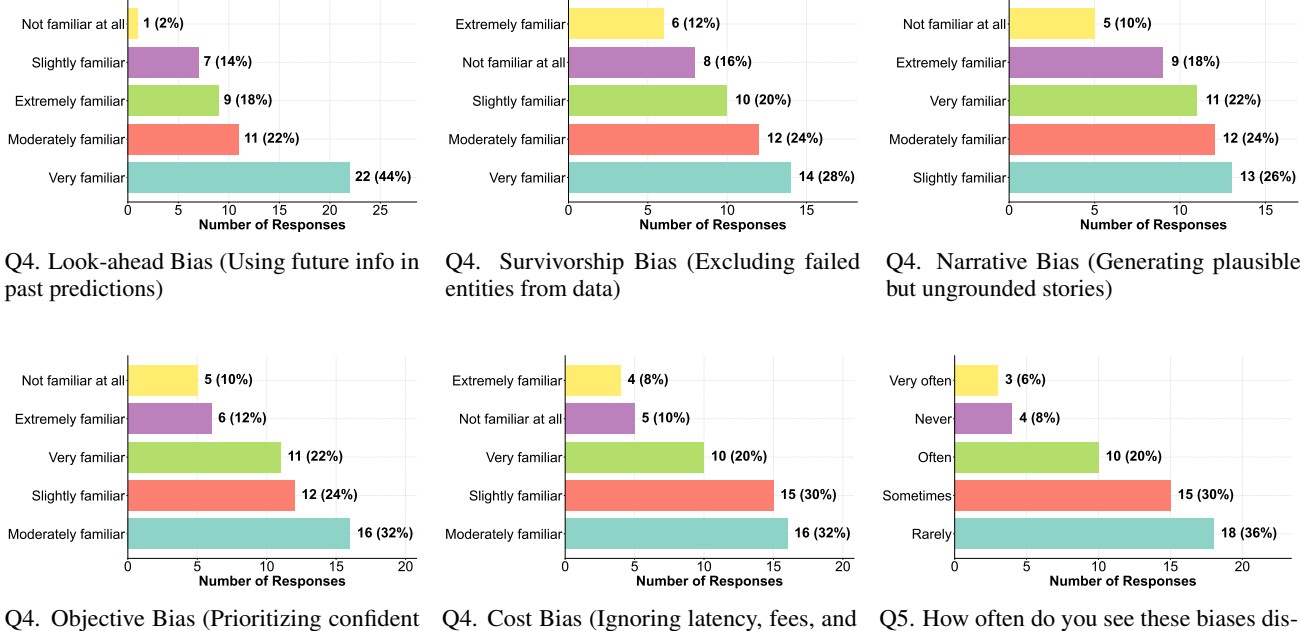

Q4. Look-ahead Bias (Using future info in past predictions)

Q4. Survivorship Bias (Excluding failed entities from data)

Q4. Narrative Bias (Generating plausible but ungrounded stories)

Q4. Objective Bias (Prioritizing confident answers over uncertainty)

Q4. Cost Bias (Ignoring latency, fees, and operational costs)

Q5. How often do you see these biases discussed in financial LLM papers/reports?

*Figure 7.* User Study - Bias Awareness and Familiarity (Total Response: 50).
Q4. Before this survey, how familiar were you with the following biases that affect financial modeling and LLM evaluation?
Q5. How often do you see these biases discussed in financial LLM papers/reports?

### A.3. Perceived Criticality of Biases

**Question 6: Criticality Assessment** This question assessed participants' opinions on how critical each bias is for the reliability of LLMs in real-world financial decision-making. Figures 8 present the results.

(1) **Look-ahead Bias:** Figure 8 shows that Look-ahead bias was perceived as the most critical, with 54% (27 participants) rating it as "Extremely critical" and 16% (8 participants) rating it as "Very critical". Only 4% (2 participants) considered it "Not at all critical", demonstrating strong consensus on the importance of temporal validity in financial LLM evaluation.

(2) **Survivorship Bias:** As illustrated in Figure 8, Survivorship bias was also perceived as highly critical, with 32% (16 participants) rating it as "Very critical" and 28% (14 participants) as "Extremely critical". The combined 60% rating it as either "Very" or "Extremely critical" indicates strong recognition of the importance of including failed entities in evaluation datasets.

(3) **Narrative Bias:** Figure 8 shows that Narrative bias was perceived as critical by most participants, with 40% (20 participants) rating it as "Very critical" and 20% (10 participants) as "Extremely critical". The high criticality ratings (60% combined) suggest awareness of the risks posed by plausible but ungrounded explanations generated by LLMs.

(4) **Objective Bias:** As shown in Figure 8, Objective bias received moderate-to-high criticality ratings, with 44% (22 participants) rating it as "Very critical" and 36% (18 participants) as "Moderately critical". The lower proportion rating it as "Extremely critical" (10%, 5 participants) compared to other biases may reflect less awareness of uncertainty quantification issues.

(5) **Cost Bias:** Figure 8 indicates that Cost bias had the most distributed criticality ratings, with 34% (17 participants) rating it as "Very critical" and 28% (14 participants) as "Moderately critical". However, 18% (9 participants) rated it as only "Slightly critical", suggesting that operational cost considerations may be undervalued in current evaluation practices.

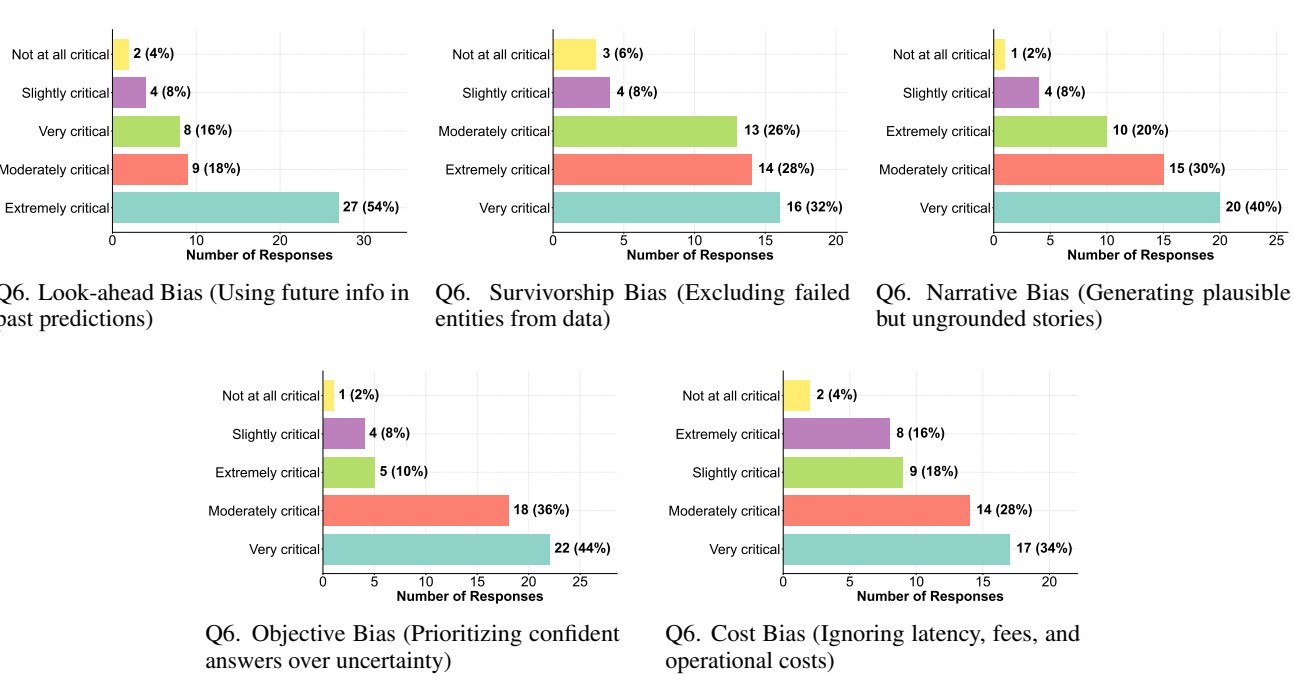

Q6. Look-ahead Bias (Using future info in past predictions)

Q6. Survivorship Bias (Excluding failed entities from data)

Q6. Narrative Bias (Generating plausible but ungrounded stories)

Q6. Objective Bias (Prioritizing confident answers over uncertainty)

Q6. Cost Bias (Ignoring latency, fees, and operational costs)

*Figure 8.* User Study - Perceived Criticality of Biases (Total Response: 50).
Q6. In your opinion, how critical is the impact of each specific bias on the reliability of LLMs in real-world financial decision-making?

## A.4. Evaluation Practices and Tool Availability

**Question 7: Confidence in Benchmark Models** Figure 9 (Q7) shows participants' confidence that benchmark models capture real-world financial scenarios, including latency, costs, and realistic dataset availability. The results reveal significant skepticism: 36% (18 participants) reported being only "Slightly confident", and 26% (13 participants) were "Not confident at all". Only 12% (6 participants) were "Very confident", and merely 4% (2 participants) were "Extremely confident". This finding highlights widespread concern about the realism and comprehensiveness of current evaluation benchmarks.

**Question 8: Guidance from Current Research** Figure 9 (Q8) presents participants' assessment of whether current LLM-in-finance research provides clear guidance for identifying or measuring biases. The results are striking: 48% (24 participants) reported that current research provides guidance only "To a small extent", and 14% (7 participants) reported "Not at all". Only 10% (5 participants) reported "To a large extent". This finding directly supports our paper's contribution, demonstrating that the field lacks standardized evaluation frameworks and clear methodological guidance.

**Question 9: Availability of Evaluation Tools** Figure 9 (Q9) shows the availability of ready-to-use tools or frameworks to evaluate biases. The results reveal a critical gap: 48% (24 participants) reported that tools are "Scarce (I have to build custom ad-hoc scripts)", and 28% (14 participants) reported that tools are "Non-existent (I have no tools for this)". Only 4% (2 participants) reported that tools are "Good (Reliable tools are available)". The combined 76% reporting scarce or non-existent tools underscores the urgent need for standardized evaluation frameworks and tools.

**Question 10: Technical Checks for Biases** Figure 9 (Q10) illustrates how often participants run specific technical checks to catch biases. The results show inconsistent practices: 36% (18 participants) run checks "Sometimes", 30% (15 participants) run them "Rarely", and 12% (6 participants) "Never" run such checks. Only 10% (5 participants) reported "Always" running checks, and 12% (6 participants) run them "Often". This finding indicates that even when practitioners are aware of biases, systematic evaluation practices are not consistently implemented.

### A.5. Barriers to Bias Mitigation

**Question 11: Biggest Bottleneck** Figure 9 (Q11) presents the most critical finding: participants' identification of the biggest bottleneck to effective bias mitigation. Half of all respondents (50%, 25 participants) identified "Lack of evaluation tools/frameworks" as the primary bottleneck, directly validating the need for our proposed structural validity framework. The second most cited bottleneck was "Difficulty in accessing high-quality, point-in-time data" (28%, 14 participants), highlighting data availability challenges. "Lack of awareness" was cited by 14% (7 participants), while "High data cost" was cited by 8% (4 participants).

**Question 12 (Optional): Most Important Missing Evaluation Guideline, Tool, or Practice** We asked participants to identify the most important missing evaluation guideline, tool, or practice in an open-ended format. Six participants provided responses, which can be found at https://github.com/Eleanorkong/Awesome-Financial-LLM-Bias-Mitigation. We visualized them as a word cloud in Figure 9 (Q12). The responses highlight several key themes: the need for long-term backtesting (5+ years), systematic bias audit frameworks that quantify bias impact, counterfactual and temporal evaluation practices, guidelines for realistic experimental settings, and robustness testing on unseen data with model explainability.

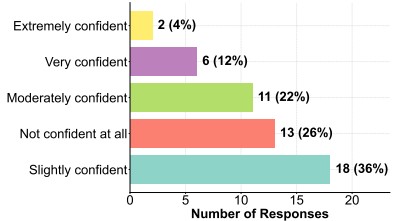

Q7. How confident are you that benchmark models capture real-world financial scenarios, including considerations of latency, costs, and the availability of realistic financial dataset?

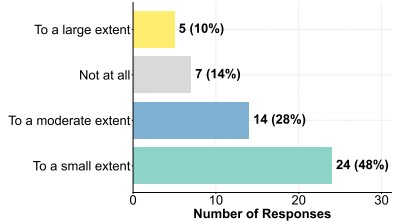

Q8. To what extent does current LLM-in-finance research provide clear guidance for identifying or measuring above biases?

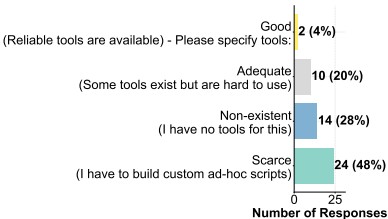

Q9. Are there ready-to-use tools or frameworks to evaluate these biases?

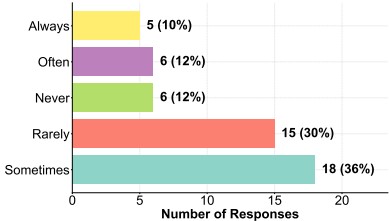

Q10. Do you run specific technical checks to catch these biases?

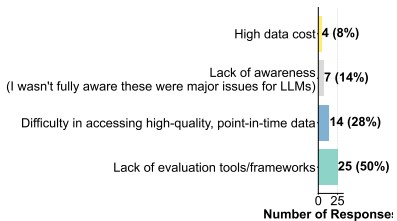

Q11. What is the biggest bottleneck to effective bias mitigation?

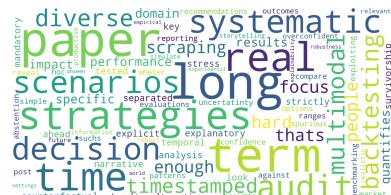

Q12. (Optional) What is the most important missing evaluation guideline, tool, or practice that would help you address financial biases in LLM-based financial systems?

*Figure 9.* User Study - Evaluation Practices and Tool Availability (Q7-Q10); Barriers to Bias Mitigation (Q11-Q12) (Total Response: 50).

### A.6. Key Findings and Implications

The user study reveals several critical insights.

**First**, while participants show varying levels of familiarity with different biases (especially look-ahead bias), these biases are rarely discussed in financial LLM papers, creating a disconnect between practitioner awareness and research attention.

**Second**, participants consistently rated the biases as highly critical for real-world reliability, particularly Look-ahead and Survivorship biases, yet technical checks and evaluation practices remain inconsistent.

**Third**, the results highlight a clear lack of standardized frameworks: 76% of participants report that evaluation tools are scarce or non-existent, and 48% report that current research provides guidance only to a small extent for identifying or measuring biases.

**Fourth**, half of all participants identified the lack of evaluation tools/frameworks as the biggest bottleneck to effective bias mitigation, directly supporting the need for our proposed structural validity framework.

**Finally**, confidence in current benchmarks is low, with 62% of participants reporting low confidence that benchmarks capture real-world scenarios, including latency, costs, and realistic dataset availability.

**Together, these findings demonstrate that the absence of standardized evaluation frameworks is a practical barrier preventing rigorous evaluation in the field, underscoring the urgent need for a structural validity framework.**

## B. Related Works

We conducted a comprehensive review and investigation of trends in LLM-for-Finance research spanning the period from 2023 to 2025. Our literature search encompassed papers from leading venues across multiple research communities: Top Machine Learning conferences (ICML, ICLR, NeurIPS), Data Mining and AI conferences (KDD, AAAI, IJCAI, CIKM), Natural Language Processing conferences (ACL, EMNLP, NAACL), the financial AI conference (ICAIF), and various workshops. All papers included in our analysis underwent human verification to ensure accuracy and relevance. The temporal trends in Financial LLM research are illustrated in Figure 1.

To facilitate systematic analysis, we employed a rule-based categorization approach using keyword matching to classify papers into distinct research categories. Our categorization framework identified the following research areas: Dataset & Benchmark (81 papers, 20.8%), Financial Reasoning (75 papers, 19.2%), Agent (54 papers, 13.8%), Financial Document (32 papers, 8.2%), Prediction & Forecasting (31 papers, 7.9%), Sentiment Analysis (29 papers, 7.4%), Financial LLM (21 papers, 5.4%), Behavior & Bias (20 papers, 5.1%), Simulation (19 papers, 4.9%), Portfolio (13 papers, 3.3%), Risk Management (11 papers, 2.8%), and Embedding (4 papers, 1.0%). All categorizations were human-verified to ensure accuracy and consistency.

We note that 226 papers (58.0% of the total corpus) were published in workshop proceedings, while 164 papers (42.0%) were published in main conference venues. Among these, we identified 5 papers that were published in both workshop proceedings and subsequently in main conference venues. While this represents duplication, we have included both versions in our analysis as they reflect distinct temporal trends and research trajectories at different stages of the publication pipeline. Additionally, we acknowledge that 27 workshop papers (11.9% of all workshop papers) currently lack associated hyperlinks, as these papers were not available online at the time of our search. We plan to include links once they are publicly released. For our bias analysis, we focused on 164 main conference papers from leading venues. The results of our bias investigation are presented in Figure 2.

All research papers included in our analysis are incorporated into an interactive dashboard designed to facilitate exploration and interaction for researchers. The dashboard enables users to filter, search, and analyze papers based on various attributes including venue, category and year. We provide an illustrative overview in Figure 10.

Below, we summarize the latest work in three key areas: (1) Bias in LLMs, (2) Bias in Financial LLMs, and (3) LLM Applications in Finance.

### B.1. Bias in LLMs

LLMs exhibit cognitive biases similar to humans during evaluation and reasoning processes. Wang et al. (2024a) identified position bias, where the order of candidate responses influences evaluation outcomes, while Wang et al. (2024b) explained

this as a consequence of the models' causal attention mechanisms and position embeddings. Furthermore, models display choice-supportive bias, maintaining their stance despite contradictory evidence due to overconfidence in initial judgments Kumaran et al. (2025). This tendency leads agents to commit attribution errors or even generate contextual hallucinations to rationalize their choices Zhuang et al. (2025). Additionally, Stureborg et al. (2024) reported that models demonstrate familiarity bias, preferring text with lower perplexity, and exhibit anchoring effects in multi-attribute judgments.

Beyond cognitive dimensions, social and gender biases inherent in training data pose significant challenges. Hu et al. (2025) demonstrated through extensive analysis that LLMs exhibit ingroup solidarity and outgroup hostility across various social groups, including gender, race, and political affiliation. Their study indicates that models replicate conflict structures regarding gender and intergroup dynamics in a manner consistent with Social Identity Theory. Notably, such hostility can be amplified when models are fine-tuned on biased data, underscoring the critical importance of data curation for building fair and robust models. Relatedly, Kotek et al. (2023) find that LLMs are 3–6 times more likely to choose an occupation that aligns with gender stereotypes, and they often ignore structural ambiguity unless explicitly prompted. They also report that the models frequently provide confident explanations that are factually inaccurate, effectively rationalizing biased outputs rather than revealing the true basis for the prediction.

### B.2. Bias in Financial LLMs

Prior work on bias in financial LLMs can broadly fall into two directions: (i) studies that identify and characterize bias phenomena in financial decision-making settings, and (ii) studies that propose mitigation methods or evaluation frameworks specifically for financial tasks.

**Bias Identification and Characterization.** Several studies have systematically identified and analyzed specific biases in financial LLM applications. Our bias analysis of 164 main conference papers revealed the presence of five key bias types: look-ahead bias, survivorship bias, narrative bias, objective bias, and cost bias. Notable works in this category include Lee et al. (2025b), which highlights survivorship bias in LLM-based investment analysis, and Dimino et al. (2025a), which examines how positional information affects financial decision-making. Zhou et al. (2025) investigates behavioral biases in LLM investment decisions, while Mehrotra et al. (2025a) studies objective bias and broader evaluation considerations. Other identification-oriented studies include Obaid & Pukthuanthong (2024), Dimino et al. (2025b), and Lakkaraju et al. (2023).

**Bias Mitigation and Solutions.** A growing body of research has proposed solutions to address identified biases. However, their coverage is uneven across bias types. For example, several works directly target look-ahead bias through model design and procedural constraints (Merchant & Levy, 2025; Kakhbod & Li, 2025). Other work emphasizes evaluation and governance of mitigation mechanisms, including robustness checks and the trade-off between safety guardrails and helpfulness (Mehrotra et al., 2025b). Complementary contributions develop broader evaluation frameworks for bias-sensitive decision settings (Vidler & Walsh, 2025) and explore reasoning-based intervention strategies (Liu et al., 2024). More generally, bias mitigation in finance often requires aligning the intervention with the data-generating process and evaluation protocol, since biases such as survivorship and look-ahead are properties of temporal data access and dataset construction rather than purely linguistic artifacts.

From researching the literature, we noticed that while solutions exist for some bias types (particularly look-ahead bias), the field is still developing comprehensive mitigation strategies for survivorship, narrative, objective, and cost biases. These results motivate our focus on transparent reporting and systematic, human-verified assessment protocols for bias considerations in financial LLM research.

### B.3. LLM Applications in Finance

LLMs are rapidly expanding their utility in finance, moving beyond text processing to complex tasks such as market simulation, quantitative research automation, and financial time series modeling. This section reviews recent developments in these areas based on contemporary research.

First, LLMs are being utilized to enhance market simulation and event analysis. Regarding Agent-Based Simulation (ABS), Yang et al. (2025b) introduced TwinMarket, where LLM-based agents utilize a Belief-Desire-Intention (BDI) framework to interpret social signals and execute trades, successfully reproducing macro-level stylized facts like volatility clustering. Complementing this, in the domain of event impact analysis, Xu et al. (2025) proposed FinRipple, a framework that aligns LLMs with market dynamics to predict ripple effects. By integrating time-varying Knowledge Graphs (KGs) and employing Reinforcement Learning guided by asset pricing theory, FinRipple effectively captures complex inter-company propagations

that traditional methods often miss.

Second, significant progress has been made in Automated Quantitative Research, particularly in addressing the challenge of alpha decay. Li et al. (2025b) proposed R&D-Agent(Q), a data-centric multi-agent framework designed to automate the full stack of quantitative strategy development, utilizing a Co-STEER agent for factor-model co-optimization. Furthermore, to specifically counteract factor decay, Tang et al. (2025a) introduced AlphaAgent, an autonomous framework that employs regularization mechanisms, including originality enforcement via Abstract Syntax Trees (AST) and hypothesis alignment. This approach ensures the generation of decay-resistant alpha factors, demonstrating robust performance across diverse market regimes.

Third, researchers are developing Financial Time Series Foundation Models to address the specific characteristics of financial data. Shi et al. (2025) presented Kronos, a foundation model pre-trained on over 12 billion financial records. Kronos employs a specialized tokenizer to convert continuous K-line (candlestick) data into discrete tokens, preserving both price dynamics and trading activity, and has demonstrated state-of-the-art performance in various downstream tasks.

Finally, rigorous Benchmarking is essential for evaluating the specialized capabilities of LLMs. Karger et al. (2024) introduced ForecastBench to evaluate forecasting capabilities while avoiding data leakage, revealing that LLMs still lag behind expert Superforecasters. Addressing the need for precise numerical reasoning, Tang et al. (2025b) developed FinanceReasoning, a comprehensive benchmark comprising over 2,200 problems and a library of 3,133 financial functions. Their evaluation highlights that while Large Reasoning Models like OpenAI o1 excel when using Program-of-Thought prompting, they still face challenges in applying correct formulas and achieving numerical precision without external tools.

## C. Template For Checklist

The checklist template is a reporting artifact that turns the Structural Validity Framework into a document a reader can audit. We present the structure of this template in Figure 11-12. It makes the evaluation assumptions visible so that a strong backtest is not accepted on trust alone. The checklist is easily fillable and accessible via https://github.com/Eleanorkong/Awesome-Financial-LLM-Bias-Mitigation. The hosted version offers the most interactive experience and supports direct copy and paste of the completed text into an appendix.

The template has five sections that match the framework and map directly onto the five sins in the methodology. Temporal sanitation targets look-ahead bias and asks for evidence that every prompt, retrieved document, tool output, and label existed at the decision time. Dynamic universe construction targets survivorship bias and asks for a time indexed tradable universe that includes later delisted entities while they are observable. Epistemic calibration targets objective bias and asks whether uncertainty and abstention are part of the interface and the score. Rationale robustness targets narrative bias and asks whether claims in explanations trace to stored sources and whether violations are measured in aggregate. Realistic implementation constraints target cost bias and asks whether latency, trading frictions, and operating costs enter the primary metric.

We propose that researchers attach the completed checklist to the appendix of their work. This practice creates a transparent audit trail for reviewers and readers. It ensures that reported improvements result from better models rather than looser evaluation standards. By standardizing these disclosures, the community can move toward a more credible and reproducible financial AI literature.

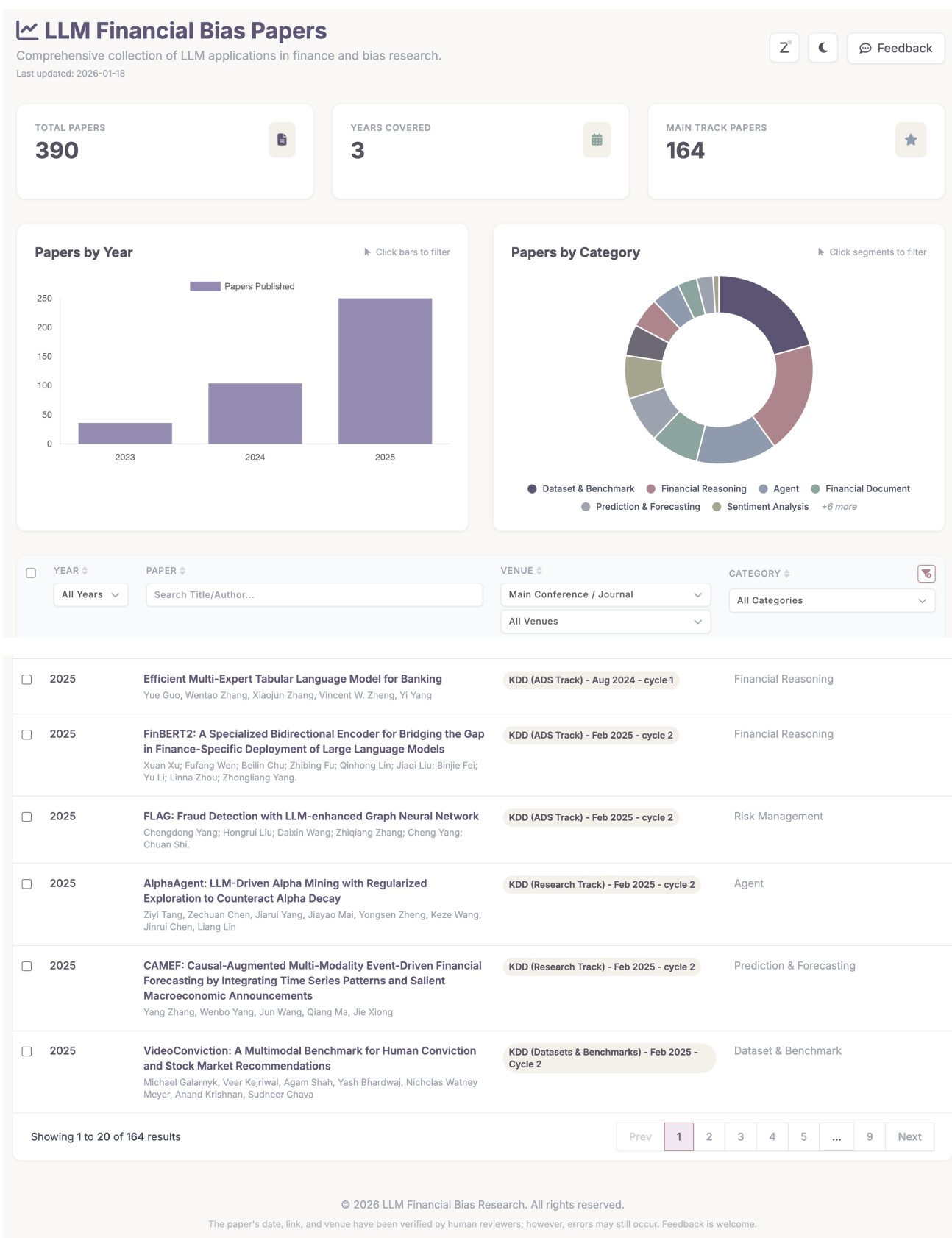

*Figure 10.* **Interactive Dashboard Overview.** Top and bottom views of the dashboard used to explore the analyzed research papers.

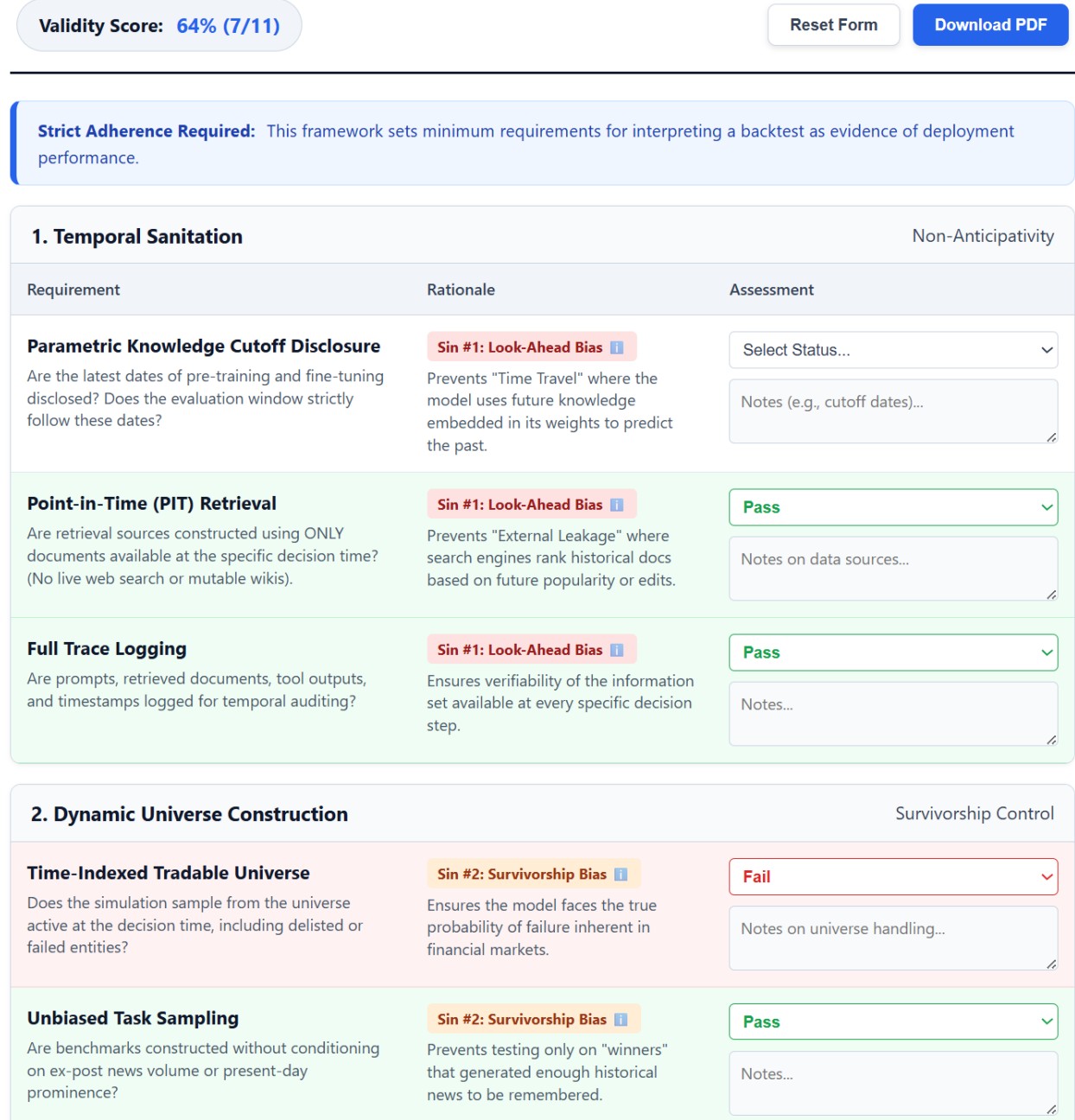

*Figure 11.* **The Structural Validity Checklist Template (Part 1).** This document maps the five potential biases to specific validation requirements to ensure reproducibility and fair comparison.

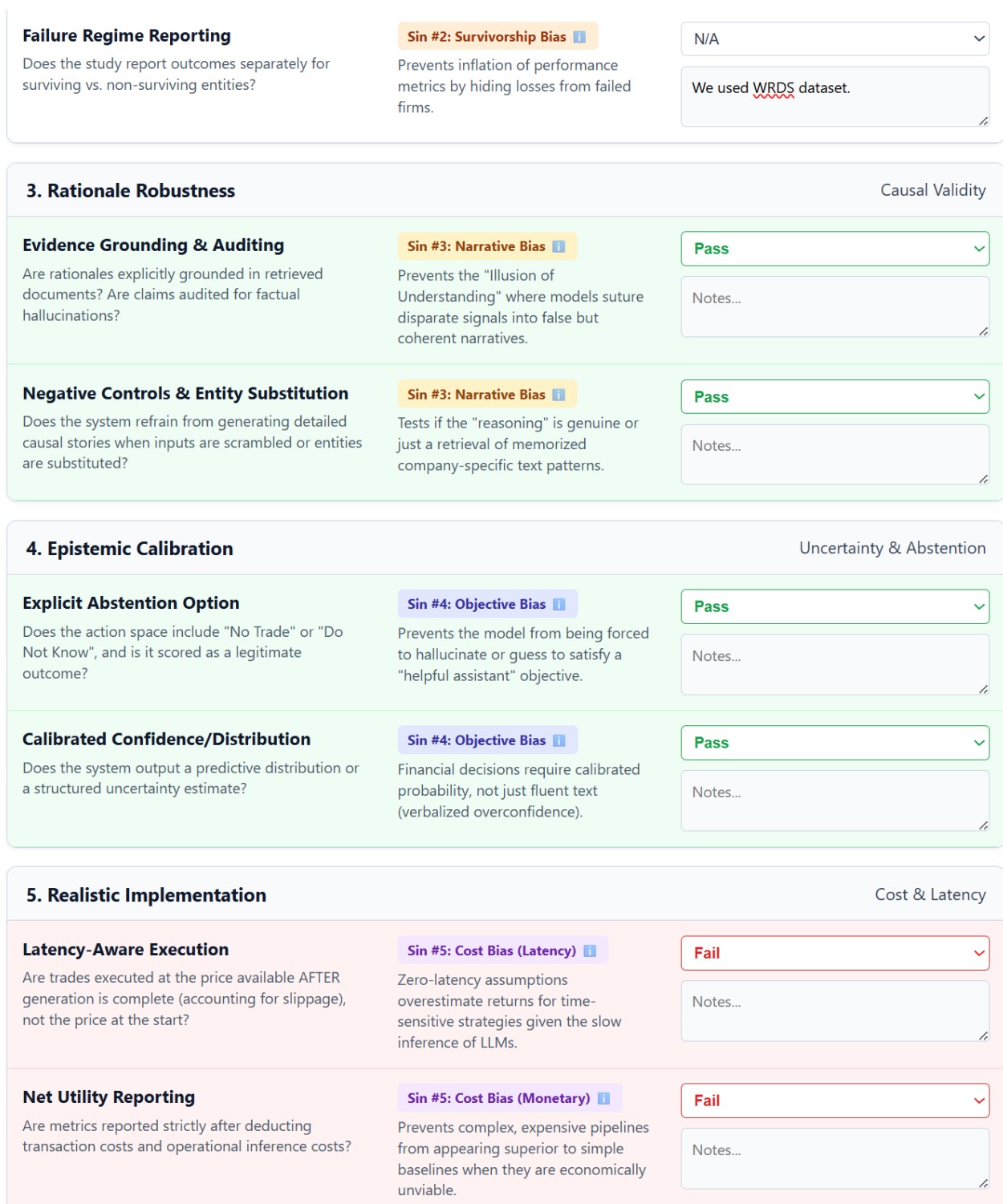

*Figure 12.* **The Structural Validity Checklist Template (Part 2).** This document maps the five potential biases to specific validation requirements to ensure reproducibility and fair comparison.

