# OpenReview forum: "Position: Evaluating LLMs in Finance Requires Explicit Bias Consideration"
_ICML.cc/2026/Position_Paper_Track — ICML 2026 Position Paper Track regular_

### Official Review · Reviewer_QrFZ · 2026-03-04

**Significance:** 3
**Argument Clarity:** 4
**Rating:** 5
**Confidence:** 4

**Questions:**

- For Sin 3, Issue 2, I didn't really understand the argumentation here regarding chain-of-thought. Are you simply talking about distribution shifts and mismatch with internal reasoning?
- I'm wondering about the practical usefulness of such checks. How realistic is it that researchers (or practitioners) can ensure all these biases are addressed when they are leveraging API-accessed, closed-source frontier models, or even open-sourced but pre-trained models, perhaps with only partial information on the training protocol? Certainly many researchers would not bother retraining a large-sized LLM from scratch just to adhere to such bias concerns, unless relevant to their study. This is not a weakness of the position or proposal per se, but is it really realistic?
- Do you have any ideas on why this gap between bias awareness and their acknowledgement (or mitigation) in papers exists? Is it a mismatch of research communities (finance vs. ML crowd), practical limitations in information access (as mentioned in the alternating views), an incentive to hide research limitations, sheer laziness, ...?
- In the introduction you state "the share of respondents using generative AI rose from 40% in 2023 to 52% in 2024". The next sentence states "It also reports that generative AI is now used or assessed by over half of respondents". Sure, since the fraction is 52% this means over half are using AI; it seems to me you are repeating the same statement twice, once with a percentage and once with words.

**Alternative Views Section:**

Yes

**Compliance With Llm Reviewing Policy A Conservative:**

Affirmed.

**Discussion Potential:**

3

**Final Justification:**

See rebuttal response

**Paper Summary:**

This paper stresses the need for more explicit and cautious bias considerations when employing LLMs in financial workflows and resulting claims. A survey of recent LLM-in-Finance papers stresses low discussion of biases, prompting the need to highlight them more explicitly. The paper discusses 5 biases ("sins") and illustrates different ways in which these biases may creep into modeling or testing workflows. A user study highlighting lacking unified evaluation as a practical hurdle is taken as a motivator to design a bias checklist covering each bias, with verifiers for correct handling. The checklist is also rendered as an interactive component.

**Position:**

Yes

**Position In Title:**

Yes

**Related Work:**

3

**Strengths And Weaknesses:**

Strenghts
- The paper is well written and makes actual surveying effort (of papers and users) to provide meaningful motivators towards their bias elaboration. I'm assuming the biases discussed are not novel by themselves, but it is nice to have a unified and engaging overview.
- The motivators naturally also result in two calls for action, and the authors put in the effort to provide meaningful, practical advice using their validity checklist rather than just pointing at some high-level "increase awareness" arguments. The resulting checklist and the discussed points are directly actionable and can be used by researchers in the field to question and challenge their experimental designs.
- The paper scopes its claims appropriately (for the most part) and roots them in references and experimental findings, rather than relying on intuitive or "commonsense" rationale arguments. This removes speculation and grounds their claims and position.

Weaknesses
- The paper focuses less on arguing *why* we need to focus on bias consideration (i.e. detailing why the position matters) and rather focuses on enumerating and illustrating the biases themselves. In some sense, it can be viewed as somewhat more of a review or taxonomy paper rather than a position paper. Nonetheless, I think it is sufficiently suitable for this paper track.
- Despite nice exposition of each bias, I would have liked each paragraph to end with a couple real, practical examples (e.g. taken from the surveyed papers) on how these biases indeed creep into existing research. For example, if a paper has failed to account for temporal coherence or uses a dataset with strong survivorship bias. This would also help provide some more references to strengthen some claims that are slightly speculative in the sense that they are simply stated as fact (e.g. "This provides a theoretical basis for why backtest returns appear inflated when real-world market drag factors are ignored."; "The smooth narratives generated by LLMs artificially flatten market complexity"; "outputs may appear confident and coherent while obscuring risk exposure or compliance violations."; "Ignoring these factors can favor heavier pipelines that score well on gross metrics but perform poorly in deployment"; "In deployment settings, accumulated $C_M$ can erode economic value.") or lack supporting evidence (the alternative view arguments, First -- Fourth). The examples in Table 1 are also a bit vague and could be made more explicit (potentially with reference).
- Relatedly, from my understanding the surveyed papers are queried for the question "do they discuss or mention such biases" but not "are they actually committing any or all of these biases"? It would be nice to also get some sense of that correspondence. This is specifically because, as the authors note themselves, there seems to be a gap between bias awareness (Fig. 7) and actual mention in research papers (and by proxy, perhaps in their "crime rate"). Does a paper acknowledge a bias but then employs other biases unknowingly? Or conversely, does it not discuss such biases but performs the correct experimental protocol? Do they correlate (discussion and failure)? This would be relevant information to better highlight the position and its significance in the context of current practice.
- I suppose some of these biases are more relevant in some studies than others. E.g. 20% of surveyed papers are reported as dataset papers; here a survivorship bias might play a big role, but look-ahead less so since no particular prediction strategy is being tested. And vice versa for a paper advocating for a new trading strategy but relying on "clean" data. It'd probably be good to also discuss a bit the key differences in terms of risk potential and occurrence of these different biases, rather than just presenting each as "one of five equal sins".
- The math notation is a bit jumbled and doesn't feel entirely unified. For example, $M$ is a financial LLM but also an information set (L91); is that the same thing? But also $I_t$ is an information set (L99). And $c_t$ is a context set but also a "retrieved information set" (L157). Samples are defined $X$ but also $X_{i,t}$ (L130), but also $x$ (L155). This is a minor point but it would be good to revise notation and ensure a consistent attribution across bias descriptions.

**Support:**

3

---

> ### Author Rebuttal · Authors · 2026-03-30
>
> We are encouraged that the reviewer finds the paper well-written, the survey effort (164 papers and 50 practitioners) meaningful, and the checklist actionable rather than merely high-level. We especially appreciate the recognition that the paper scopes its claims appropriately and grounds them in evidence, which is exactly the standard we argue the field should adopt.
>
> **[W1–W5]**
>
> While the paper describes each bias in detail, this exposition is not intended as a taxonomy for its own sake. Rather, it shows precisely how each bias changes what an experiment actually measures. Our central claim is that, in finance, standard LLM evaluation is not sufficient to support real-world claims. Some errors do not simply inflate a score slightly; they change the object of measurement itself. A backtest that uses future information no longer measures a feasible decision. A benchmark built only from surviving firms no longer reflects market reality. A result reported without costs no longer measures deployable performance. The paper therefore takes the position that structural validity checks should precede any deployment claim.
>
> To make this concrete, we tie each bias to a specific failure pattern. If a study queries a present-day search engine for historical evaluation, it introduces look-ahead contamination. If benchmark questions are generated from firms that remain prominent today, failed firms are underrepresented. If summaries are judged primarily by fluency, the model may omit weak but important warnings. If a trading system is praised for gross returns while ignoring inference cost, latency, and slippage, an uneconomic pipeline may appear successful.
>
> A key distinction is between recognizing a bias and demonstrating that it has been addressed. A paper may acknowledge look-ahead risk and still use present-day resources in historical evaluation. Conversely, another paper may never use the term explicitly, yet still avoid contamination through a strict temporal split. Our concern is therefore not whether authors name the correct biases, but whether the study meets the checklist criteria that make the evaluation valid. The checklist rewards demonstrable safeguards, not verbal awareness alone.
>
> The five biases are not equally important in every study, because their relevance depends on the task and the claim being evaluated. In dataset papers, survivorship bias is often the main concern. In trading papers, look-ahead bias, cost, and latency are often more consequential. In summarization, narrative and objective bias are typically more salient. We therefore present them as task-dependent threats to validity, while noting that a severe failure on any one dimension can already make the results hard to interpret as evidence for real financial use.
>
> To ground these points, we developed an automatic bias detection agent that analyses methodology against the checklist. Recent evidence shows that more robust evaluation can change apparently favorable conclusions. Additional examples are available in the anonymous repository. We will also revise the notation for consistency and add references to Table 1.
>
> **[Q1] CoT in Sin 3**
>
> The concern is broader than distribution shift alone. Our point is not only that CoT may fail to faithfully reflect the model’s latent decision process, but also that a fluent rationale may not be grounded in the evidence actually available at decision time. In finance, where regime shifts quickly erode the apparent causal value of polished narratives, this makes evidence-grounding and auditability essential: rationales should be grounded in temporally valid sources and fact-checked rather than treated as self-authenticating explanations.
>
> **[Q2] Practicality**
>
> The framework does not require retraining. Instead, it focuses on what remains verifiable at the system level, such as point-in-time data, retrieval traces, abstention evaluation, and cost measurement. The relevant standard is not perfect observability, but alignment between the strength of the claim and the strength of the available evidence. The framework is therefore intended as a practical guard against overclaiming, even under limited model access.
>
> **[Q3] Awareness Gap**
>
> The gap is likely structural. The ML community rewards benchmark gains; finance values implementability. Point-in-time data are costly, delisting histories hard to obtain, and validation details easy to defer. Researchers may know the problem and still fail to build it into the protocol. This is why we think the checklist really matters. It can bridge research cultures and lowers the cost of careful evaluation.
>
> **[Q4] Clarify**
>
> The two sentences report different findings: the first is the overall GenAI adoption rate (from 40% to 52%); the second is that over half of firms are deploying or evaluating specific LLM workloads such as investment research, document processing, and risk analytics. We agree the original phrasing obscured this distinction and will revise.

---

> > ### Author Rebuttal · Reviewer_QrFZ · 2026-04-01
> >
> > Thanks, your rebuttal seems to reiterate some of the points I made. So it would be nice to add your comments and address some of them in an overhauled version. E.g. the connection between bias awareness, bias mention/acknowledgement, and committing bias mistakes; the relevancy of different biases to different types of studies; results of bias detection analysis vs. awareness; and so on. These will help make your position even more practical and contribute to the field with more impact by broadening understanding and relations among the research community. I'll maintain my score.

---

### Official Review · Reviewer_nmdf · 2026-03-09

**Significance:** 4
**Argument Clarity:** 3
**Rating:** 5
**Confidence:** 4

**Questions:**

1) Can you provide 2–3 concrete case studies (e.g., trading agent / forecasting / QA) where applying the checklist changes the main conclusion (e.g., “valid” to “invalid”)?

2) How should the checklist be applied to exploratory papers that do not make deployment claims, without unfairly penalizing early-stage research?

3) For “narrative” and “objective” bias, what are the minimum recommended tests when outputs are not immediately verifiable (e.g., macro narratives, free-form rationales)?

**Alternative Views Section:**

Yes

**Compliance With Llm Reviewing Policy A Conservative:**

Affirmed.

**Discussion Potential:**

4

**Paper Summary:**

The paper argues that many evaluations of LLMs in finance are structurally invalid because they fail to control for finance-specific biases (e.g., look-ahead bias, survivorship bias, narrative bias, objective misspecification, and cost/implementation constraints). The authors support this position with a broad review of recent literature and a survey indicating that such biases are rarely addressed, making results non-comparable and often non-deployable. They propose a Structural Validity Framework and a reviewer-/author-facing pass/fail checklist to standardize evaluation and reporting for credible claims in financial ML.

**Position:**

Yes

**Position In Title:**

Yes

**Related Work:**

3

**Strengths And Weaknesses:**

**Strengths**

* Clear, actionable position with concrete outputs (framework + checklist) that reviewers and authors can adopt.

* Strong motivation grounded in well-known pitfalls of financial research (high risk of “illusory validity” in backtests).

* High potential impact on community norms for evaluation, especially as LLM use in finance grows.

**Weaknesses**

* A strict “pass/fail” framing may be too rigid for exploratory work that does not claim deployability.

* Would benefit from more task-specific operational guidance (forecasting vs trading agents vs financial QA) and a few worked “before/after” case studies showing how conclusions change under the checklist.

* Could more systematically connect to established finance evaluation standards (backtesting conventions, transaction cost modeling, etc.).

**Support:**

4

---

> ### Author Rebuttal · Authors · 2026-03-30
>
> We thank Reviewer nmdf for raising an important set of concerns!
>
> **[W1][Q2] The checklist does not penalize exploratory work.**
>
> The checklist clarifies what kind of conclusion the evidence can support. A paper can still make a valuable contribution by showing that a method looks promising in a controlled historical setting. The stronger question is whether that same evidence also supports claims about real financial use. If not, the contribution still matters, but it should be interpreted as proof of concept rather than deployment evidence. This distinction is already familiar in finance. A backtest that uses information not truly available at the time may still be useful as an idea test, but it is not treated as evidence of deployable performance [1-2]. A study built on a universe that keeps only surviving firms does not fully reflect real market risk [3]. Returns reported without realistic costs and delay may also differ materially from returns that could actually be achieved [1][4]. Failing a deployment item therefore does not mean that the paper has no value. It means only that the result should not be read as deployment evidence. If a paper makes an exploratory capability claim, then deployment-oriented items are naturally out of scope or N/A.
>
> **[W2][W3][Q1] The checklist changes the interpretation of results in trading, forecasting, and financial QA in concrete ways.**
>
> For a trading agent, the key question is simple. Could the agent really have made that decision on that date, using only the information available then, in that stock universe, after accounting for the real cost of acting. Consider a study that backtests only on firms that are still prominent today and reports a high Sharpe ratio while giving limited treatment to execution cost and delay. Before applying the checklist, the headline may read as “the model outperforms the market.” After applying the checklist, the interpretation becomes narrower. If future leakage is not ruled out, firms that disappeared are excluded, and realistic costs are missing, then the result supports only a constrained historical simulation, not evidence of deployable alpha.
>
> The same logic applies to forecasting. If a macro or earnings forecasting system relies on later-revised data series or on documents that are cleaner today than they were at the historical decision date, then the task may look like forecasting on paper while actually functioning more like reading history with hindsight. The key question is whether the input truly existed and remained accessible at that decision time. If not, the result should be interpreted as hindsight reconstruction rather than genuine real-time forecasting ability.
>
> Financial QA shows the same issue in another form. If most questions are about large firms that survived, remained well known, the model may rely more on memory and visibility than on robust financial reasoning. Once the benchmark includes delistings, bankruptcies, mergers, and mixed-evidence cases, the meaning of the score changes. Performance becomes closer to realistic financial QA, and earlier high accuracy is revealed as performance on easier cases. These examples also clarify the link to established finance standards. Temporal sanitation extends the principle that a study should use only the information available at the decision time. Dynamic universe construction extends the principle that firms that later disappear should not be silently removed [3]. Realistic implementation extends the principle that reported performance should reflect costs and delay rather than a frictionless paper setting [1][4].
>
> **[Q3] The minimum tests for narrative bias and objective bias** First, a grounding test. Each important claim should be traceable to a source passage, table, or filing that actually existed at that time. Second, a balance test. If the input contains both positive and negative evidence, the model should preserve that tension instead of turning it into one smooth story. Third, an insufficient-evidence test. If key evidence is removed, scrambled, or made contradictory, a trustworthy system should become more cautious, not more confident. Fourth, an abstention test. “I do not know” or “No trade” should be valid outputs. For these reasons, the checklist is not rigid toward exploratory work. It is a minimal scientific boundary that prevents exploratory evidence from being overstated as deployment evidence.
>
> [1] Chan, E. P. (2026). Quantitative trading: how to build your own algorithmic trading business. John Wiley & Sons.
>
> [2] Bailey, D. H., Borwein, J. M., Lopez de Prado, M., & Zhu, Q. J. (2015). The Probability of Backtest Overfitting. Journal of Computational Finance (Risk Journals).
>
> [3] Brown, S. J., Goetzmann, W., Ibbotson, R. G., & Ross, S. A. (1992). Survivorship bias in performance studies. The Review of Financial Studies, 5(4), 553-580.
>
> [4] Harvey, C. R., & Liu, Y. (2015). Backtesting. Available at SSRN 2345489.

---

> > ### Author Rebuttal · Reviewer_nmdf · 2026-04-02
> >
> > The rebuttal addresses several of my concerns. In particular, the authors clarify that the checklist is intended to scope claims (proof-of-concept vs deployment evidence) and should not penalize exploratory work, with deployment-oriented items treated as out-of-scope or N/A when appropriate. They also provide a helpful set of minimum recommended tests for narrative and objective bias (evidence grounding, balance, insufficient-evidence sensitivity, and abstention).
> >
> > However, my concern about concrete “before/after” evidence remains only partially resolved. While the rebuttal explains how the checklist would change interpretation for trading/forecasting/QA, it remains largely illustrative rather than providing 2–3 worked case studies demonstrating how applying the checklist changes the main takeaway in practice.
> >
> > Follow-up questions
> >
> > 1) Could you include at least one fully worked example (ideally 1–2 pages or an appendix): select a representative public paper/system and show the checklist applied end-to-end (what evidence is used for each check, pass/fail outcomes), and how that changes the paper’s interpreted claim (e.g., deployable alpha / proof-of-concept)?
> >
> > 2) For the proposed “minimum tests” (grounding/balance/insufficient-evidence/abstention), can you specify practical decision criteria (e.g., what constitutes a “pass,” suggested metrics or annotation protocol, and how to report violation rates) so that reviewers can apply Check 3/4 consistently?

---

### Official Review · Reviewer_j3Pm · 2026-03-11

**Significance:** 3
**Argument Clarity:** 3
**Rating:** 3
**Confidence:** 4

**Questions:**

1. **Pass/Fail Operationalization:** For each checklist item, what are the *minimum concrete pass/fail criteria* (ideally with a worked example of auditing an existing paper/system using typical public artifacts such as paper text, released code/data, and logs)?

2. **Evidence of Practical Impact:** Can you provide at least one *end-to-end case study or re-evaluation* showing how enforcing the checklist changes conclusions (e.g., absolute performance, model ranking, estimated “alpha,” or net utility) compared to common evaluation setups?

3. **Temporal Sanitation Verification (Weights + RAG):** How do you propose verifying *no look-ahead leakage* (a) in model weights—especially for closed-source models with opaque cutoffs—and (b) in retrieval from mutable sources (web pages, edited docs)? What specific tests and provenance/logging requirements do you recommend?

4. **Dynamic Universe Standard:** What is the minimal standard for “dynamic universe” across different task types (trading backtests vs QA/analysis)? What diagnostics must be reported (e.g., delisting/M&A coverage, fraction of samples from entities that delist within the window, stratified results for surviving vs non-surviving)?

5. **Uncertainty/Abstention + Cost/Latency Scoring:** When allowing abstention (“No trade/Do not know”) and accounting for cost/latency, what scoring rules and standardized reporting should be used to avoid gaming and to enable fair budget-matched comparisons?

**Alternative Views Section:**

Yes

**Compliance With Llm Reviewing Policy A Conservative:**

Affirmed.

**Discussion Potential:**

3

**Final Justification:**

The rebuttal addressed my main concerns. I raise my score.

**Paper Summary:**

This position paper argues that current evaluation practices for Large Language Models (LLMs) in finance often produce misleading results due to overlooked domain-specific biases, and that structural validity should be enforced before any results are used to support deployment claims. The authors identify five recurring sources of distortion in financial LLM applications: **look-ahead bias**, **survivorship bias**, **narrative bias**, **objective bias**, and **cost bias**. These biases can inflate reported performance, contaminate backtests, and create an illusion of effectiveness that does not translate to real-world financial decision-making. The paper supports this motivation through a review of 164 LLM-for-finance papers (2023–2025) and a user study of researchers and practitioners, highlighting the limited attention given to systematic bias diagnosis and the lack of standardized evaluation tools.

To address this gap, the paper proposes a **Structural Validity Framework** accompanied by an evaluation checklist that defines minimum requirements for credible financial LLM evaluation. The framework consists of five components: (1) **Temporal Sanitation** to prevent future information leakage, (2) **Dynamic Universe Construction** to control survivorship bias, (3) **Rationale Robustness** to ensure explanations are grounded in verifiable evidence, (4) **Epistemic Calibration** to incorporate uncertainty and allow abstention, and (5) **Realistic Implementation Constraints** to account for transaction costs, inference costs, latency, and net utility. The authors advocate that these requirements should be treated as pass–fail conditions, and that results failing any of them should be interpreted only as proof-of-concept rather than evidence of deployable performance. The paper’s central position is a call for the research community to explicitly diagnose and report structural biases and to adopt standardized validity checks to enable more reliable and reproducible evaluation of financial LLM systems.

**Position:**

Yes

**Position In Title:**

Yes

**Related Work:**

3

**Strengths And Weaknesses:**

**Strengths**

* **Strong and timely motivation.** The paper addresses an important issue: evaluation practices for financial LLM systems may produce misleading conclusions due to domain-specific biases (e.g., look-ahead leakage, survivorship effects, and frictionless assumptions). This problem is highly relevant to the ICML community, as concerns about evaluation validity and real-world claims extend beyond finance.

* **Clear conceptual structure.** The taxonomy of five biases and the corresponding **Structural Validity Framework** provide a coherent and well-organized position. The checklist formulation is practical and may help standardize reporting and encourage more rigorous evaluation practices.

* **Well-argued and well-situated.** The paper cites a broad range of recent literature and includes an “Alternative Views” section that discusses credible counterarguments, which strengthens the completeness of the position.

**Weaknesses**

* **Limited novelty.** Many identified issues (e.g., point-in-time data, survivorship control, transaction costs) are established principles in empirical finance and existing evaluation discussions. The main contribution is a synthesis rather than a fundamentally new methodological or theoretical advance.

* **Framework remains high-level and not fully operationalized.** The proposed checklist lacks precise metrics, formal criteria, or standardized procedures, making practical adoption and reproducibility potentially ambiguous.

* **Insufficient empirical validation of the central claim.** The literature survey and user study demonstrate awareness gaps but do not show that applying the proposed framework materially changes evaluation outcomes or improves validity.

* **Evaluation evidence is limited.** There is no case study or re-evaluation demonstrating the magnitude of distortion caused by each bias or the practical impact of enforcing the checklist.

**Support:**

2

---

> ### Author Rebuttal · Authors · 2026-03-30
>
> Thanks for your feedback! We address novelty, operationalization, and empirical impact below.
>
> **[W1]** Although traditional finance biases are widely recognized, they are systematically overlooked in financial language model evaluations. An analysis of 164 papers reveals poor reporting practices, with survivorship bias addressed in only 1.2 percent of studies. Furthermore, a user study confirms that 74 percent of practitioners lack adequate evaluation tools. Our research addresses this problem and offers three novel contributions. First, we unify these known biases into a framework specifically tailored for unique model failure modes, including parametric knowledge leakage, overconfidence induced by reinforcement learning, and narrative coherence that masks uncertainty. Second, we empirically document the existing evaluation gap in the current literature. Third, we provide practical solutions by introducing a concrete checklist and an automated detection agent.
>
> **[W2]** We agree that point-in-time data, survivorship control, and transaction costs are not new ideas by themselves. The contribution of our paper is not to rename those principles, but to turn them into a single minimum standard for deciding when a financial LLM result can be read as credible evidence. Existing finance guidance does not fully address future information embedded in model weights, temporal contamination from editable web sources and RAG, or persuasive rationales that go beyond the evidence. Please refer to the papers cited in our reference section (Golchin et al., 2023, Paleka et al., 2025, Turpin et al., 2023). For this reason, our checklist is not a high-level suggestion. It is a pass or fail audit rule that can be applied using public paper text, code, and logs.
>
> **[W3][W4][Q2]** The practical impact of the framework is visible in [1]. Their study re-evaluates LLM investing strategies from 2004 to 2024, over 100 plus symbols, and over historical S&P 500 universes that include delisted names. Under selective settings, the strategies can appear strong. Under broader settings, that conclusion changes. For example, the previously reported FinMem Sharpe on TSLA was 2.679, but in the same paper’s re-run of that window FinMem drops to 0.927 with GPT-4o-mini and 0.404 with GPT-4o. The broader Composite setup makes the contrast clearer. In the Volatility Effect setting, Buy and Hold reaches a Sharpe of 0.703, while FinMem is at -0.228 and FinAgent at 0.241. The paired t-tests also show that Buy and Hold significantly outperforms both LLM agents across the robust setups, and the authors report no statistically significant alpha for either LLM agent. This is exactly our point. Once the evaluation uses a broader universe and a longer horizon, the conclusion changes from apparent LLM superiority to performance driven by selective evaluation.
>
> **[Q1]** If we audit [1] using only the public paper and appendix, point-in-time external data alignment passes, because the paper states that news and filings are indexed by date and aligned to each backtest window. Dynamic universe construction also passes, because the study uses historical constituent lists and explicitly includes delisted symbols. By contrast, model-weight temporal verification does not pass for a strong deployment claim, because the paper itself states that possible leakage in proprietary pretrained models such as GPT-4o cannot be fully verified. Realistic implementation also does not fully pass under a strict rule. The appendix reports API cost, but that cost is not incorporated into the main metric. This is why the checklist is useful. It does not replace careful research design. It shows, within the same paper, which design choices support a strong claim and which choices still weaken the claim.
>
> **[Q3][Q4][Q5]** Since closed-source models inherently preclude verifying the absence of look-ahead leakage, our framework logically bounds the strength of any deployment claim to its supporting evidence. Specifically, claiming "deployable alpha" necessitates rigorous evidence, such as open weights and explicit cutoffs; otherwise, results must be interpreted strictly as proofs-of-concept. Furthermore, to ensure empirical validity, we establish a minimum dynamic-universe standard that eliminates survivorship bias by mandating the inclusion of all historical entities at time $t$ (e.g., bankruptcies and M&As) across both trading and QA tasks. Additionally, we formalize abstention as a valid strategic action rather than a default failure, requiring transparent reporting of abstention rates and cost-adjusted net utility to prevent metric gaming. Ultimately, the core contribution of this paper is not to propose a novel metric, but to define the minimum methodological conditions required to scientifically validate financial LLM results.
>
>
> [1] Li, W. W., Kim, H., Cucuringu, M., & Ma, T. (2026). Can LLM-based Financial Investing Strategies Outperform the Market in Long Run?. KDD

---

> > ### Author Rebuttal · Reviewer_j3Pm · 2026-04-04
> >
> > Thank you for addressing my concerns. I am now satisfied with the response and will update my score to reflect this.

---

### Official Review · Reviewer_tQNT · 2026-03-13

**Significance:** 4
**Argument Clarity:** 3
**Rating:** 5
**Confidence:** 4

**Questions:**

1. Have the authors considered how parts of the Structural Validity Checklist might be automated? For instance, could an "LLM-Evaluator" be designed to detect "Narrative Bias" by automatically generating the entity-swap prompts you described?
2. In Check 4, you suggest that "Abstaining" should be considered a correct result. How do we prevent researchers from gaming this by creating models that abstain on all difficult cases to inflate accuracy/precision while having near-zero recall? Should the framework mandate a risk-adjusted or utility-based metric instead?

**Alternative Views Section:**

Yes

**Compliance With Llm Reviewing Policy A Conservative:**

Affirmed.

**Discussion Potential:**

3

**Paper Summary:**

This position paper addresses a critical yet often overlooked gap in the evaluation of Large Language Models (LLMs) for financial applications: the lack of domain-specific structural validity. The authors argue that high performance on standard NLP benchmarks often masks fundamental flaws—labeled as "five recurring sins"—that render models useless or even dangerous for real-world deployment. By proposing a "Structural Validity Framework" and an actionable checklist, the paper moves the conversation from "Does it work on this dataset?" to "Is the evaluation itself valid for finance?". This is a timely, well-organized, and much-needed intervention in the FinLLM space.

**Position:**

Yes

**Position In Title:**

Yes

**Related Work:**

3

**Strengths And Weaknesses:**

Strengths：
1. The paper doesn't just describe technical research; it takes a firm stance against the "leaderboard-chasing" culture in FinLLM research that ignores the structural nuances of financial data.
2. The review of 164 papers provides significant evidence that the community is currently struggling with these biases, making the "Call to Action" feel both urgent and justified.
3. The "Five Sins" framework (Look-ahead, Survivorship, Narrative, Objective, and Cost biases) provides a shared vocabulary that will likely spark constructive debate at ICML regarding how we define "progress" in specialized AI.

Weaknesses:
1. The paper identifies "Objective Bias" (overconfidence/unwillingness to abstain) as a financial sin. However, this is a well-documented general LLM flaw (calibration issues). While the authors correctly state that the consequences in finance are higher, the paper could do more to delineate what makes this specifically a finance bias versus a general model limitation that finance researchers must inherit.
2. Among the five sins, "Narrative Bias" feels the most subjective. While the "entity-swap test" is a clever idea, the boundary between "robust causal reasoning" and "hallucinated narrative" remains fuzzy. I am concerned that without more automated or objective metrics for Check 3, this part of the checklist might lead to inconsistent peer reviews.
3. The position paper focuses primarily on static backtesting validity. However, in financial markets, the act of deployment often changes the environment (Alpha Decay). Addressing how structural validity handles the "reflexivity" of markets would have added another layer of depth to the significance of the position.

**Support:**

3

---

> ### Author Rebuttal · Authors · 2026-03-30
>
> We thank Reviewer tQNT for the positive assessment and thoughtful questions. We are encouraged that the reviewer recognises the urgency of moving beyond leaderboard-chasing in FinLLM research, and that our Five Sins framework and 164-paper review provide both a shared vocabulary and significant evidence for the community.
>
> **[W1] Objective Bias in Financial LLM Evaluation**
>
> This is a fair distinction. We do not claim that overconfidence or miscalibration is unique to finance. Our point is that, in finance, the evaluation target is materially different from generic LLM helpfulness: financial deployment requires calibrated uncertainty, abstention when evidence is weak, and adherence to risk and compliance constraints. In this sense, “objective bias” is finance-specific at the level of evaluation rather than mechanism. A model may perform well on generic accuracy or preference-based metrics while still being misaligned with the requirements of responsible financial use.
>
> **[W2] Narrative Bias Operationalisation**
>
> We agree that “Narrative Bias” can sound subjective if interpreted as judging whether a rationale merely appears coherent or causal. However, this is not the intended interpretation of Check 3. In our checklist, we operationalize this section as Evidence Grounding & Auditing: whether rationales are explicitly tied to retrieved documents, whether claims are checked for factual hallucinations, and whether the system exposes implementable mechanisms such as citation extraction, grounding verification, or fact-checking pipelines. Under this interpretation, the goal is not to subjectively assess narrative plausibility, but to evaluate whether generated rationales are traceable and auditable.
>
> **[W3] Alpha Decay and Reflexivity**
>
> In real markets, deployment can change the environment. Other traders respond, similar signals get crowded, and alpha can fade. Your comment about reflexivity and alpha decay is therefore highly important. But it concerns a later stage of the problem. It is about a real signal becoming weaker once it enters the market. Structural validity deals with an earlier problem, where an experiment can create the appearance of a signal that was never real in the first place. Unless that earlier problem is removed, it is impossible to tell whether later disappointment comes from market adaptation or from a flawed evaluation from the start. In that sense, reflexivity does not weaken the paper’s position. It explains why this first layer of discipline is necessary. The framework already moves in that direction through latency, slippage, and operating cost, which capture the most direct ways in which a signal loses value in real use. The intended claim is therefore modest but important. Structural validity is not a full guarantee of deployable alpha. It is the minimum condition that must be met before any deployment claim can be taken seriously.
>
> **[Q1] Automating the Checklist**
>
> Yes. We have built a prototype automatic bias detection agent (in our repository) that accepts a paper PDF, audits both the paper text and released code, and outputs structured Pass/Fail/N/A labels for each checklist item with actionable recommendations. It probes post-cutoff knowledge for temporal sanitation and flags missing inference cost discussion for cost bias. We provide several case studies in our anonymous repository.
>
> **[Q2] Treating Abstention**
>
> Treating abstention as a valid outcome does not mean rewarding a model for avoiding difficult cases. The point is that in finance, not acting can be the correct action. A model that says “I do not know” or “No Trade” in the right situations may be behaving more safely than a model that is forced to guess. At the same time, a model that abstains on every hard case has not solved the task. It has shown that it cannot act when it matters. Such a model would fail once the evaluation includes coverage, recall, missed opportunity, and net utility. For that reason, abstention should never be read on its own. It should be judged together with how often the model acts and what value those actions create. A better evaluation looks at whether the model keeps useful performance at meaningful coverage, whether actions are good when it does act, whether overconfidence is reduced, and whether “No Trade” beats simple alternatives such as staying in cash. In trading, abstention belongs in the action space because no trade is a real decision. But it only has value when it improves utility under realistic costs. The purpose of the framework is therefore not to inflate accuracy through abstention. It is to prevent the evaluation from forcing false certainty and unnecessary trades.

---

> > ### Author Rebuttal · Reviewer_tQNT · 2026-04-03
> >
> > I already gave a positive rating for this work. There is no need more comments for it.

---

### Decision · Program_Chairs · 2026-04-30

**Decision:**

Accept (regular)

**Comment:**

The authors take the position that evaluating LLMs in finance requires explicit consideration of bias.

The reviewers agreed that the paper has several strong points:

* The paper takes a firm stance against the “leaderboard-chasing” culture in FinLLM research that ignores the structural nuances of financial data.

*The paper is well written. The review of 164 papers provides strong evidence that the community is currently struggling with these biases.

* The “Five Sins” framework (look-ahead, survivorship, narrative, objective, and cost biases) provides a shared vocabulary that will likely spark constructive debate at ICML.

* The paper presents a clear, actionable position with concrete outputs (framework and checklist).

* The authors addressed the main concerns and questions raised by the reviewers during the rebuttal.

The AC has read the authors’ rebuttals and comments and has incorporated them into the decision-making process.